# HYPERGRAPH-NATIVE MESSAGE PASSING: AN INCIDENCE-CENTRIC PERSPECTIVE

## ABSTRACT

While hypergraphs encapsulate higher-order interactions among entities and transcend the pairwise connections characteristic of traditional graphs, their prevailing learning approaches predominantly inherit from graph neural networks, adhering to the established message passing paradigm. These methods frequently conceptualizes hyperedges as special nodes, facilitating the transmission of aggregated messages through hyperedges instead of direct messages between adjacent nodes. Such a paradigm is prone to information loss, especially in the context of large hyperedges that bridge a heterophilic array of nodes. To mitigate this shortcoming and enhance high-order message passing, we propose the Hypergraph-native Message Passing (HMP) framework, which leverages full-rank interactions among the incidences along the underlying hypergraph and its dual. In contrast to the conventional node-centric approaches, this incidence-centric perspective adeptly manages incidence-level tasks, such as hyperedge-dependent labelling, and seamlessly integrates virtual incidences for both hyperedge- and node-level tasks. Empirical evaluations demonstrate that HMP achieves a substantial improvement over state-of-the-art methods on 6 hyperedge-dependent labelling benchmarks, with an increase in accuracy ranging from 2.3% to 28.9%, while also delivering competitive results on 13 node classification benchmarks. Code to reproduce all our experiments is available at `https://anonymous.4open.science/r/HMP-FB14/`.

## 1 INTRODUCTION

Although graphs have emerged as an indispensable instrument for the modelling of complex systems, in which nodes symbolize entities and edges delineate pairwise relationships (Wu et al., 2020), numerous real-world systems manifest higher-order interactions that fall beyond the descriptive capacity of conventional pairwise graph models (Battiston et al., 2020). This deficiency has spurred the investigation into hypergraphs, a conceptual extension of graphs that provides a more nuanced structural representation by enabling edges, referred to as hyperedges, to affiliate more than two nodes, or termed vertices, thereby directly encoding intricate and multipartite relationships (Antelmi et al., 2023), as exemplified in Figure 1a. Hypergraphs prove particularly efficacious in contexts where multi-dimensional interactions are widespread, such as co-authorship networks (Bai et al., 2021) or multi-agent systems (Zhang et al., 2022), while also remaining congruent with scenarios where traditional graphs hold sway, including social networks (Yu et al., 2021), biological systems (Gopalakrishnan et al., 2022; Zhang et al., 2024; Xu et al., 2022), and knowledge graphs (Liu et al., 2023a).

Despite the capability of hypergraphs to encapsulate complex interactions, the evolution of hypergraph neural networks (HNNs) has been substantially guided by the methodologies developed for their graph-based counterparts (Feng et al., 2019; Yadati et al., 2019; Bai et al., 2021). Contemporary HNNs predominantly adhere to the message passing schema of graph neural networks, where hyperedges are conceptualized as special nodes through which message aggregation occurs (Dong et al., 2020; Huang & Yang, 2021; Georgiev et al., 2022), as illustrated in Figure 1b. For instance, AllSet (Chien et al., 2021) constructs its layers by employing two multiset functions: one for aggregating messages from nodes to hyperedges, and another from hyperedges to nodes. While this approach is conceptually straightforward, it is not without its drawbacks; a notable limitation is the potential for information loss, as the messages that nodes collect from their neighbours are effectively squashed through hyperedges (Di Giovanni et al., 2023). This squashing undermines the high-

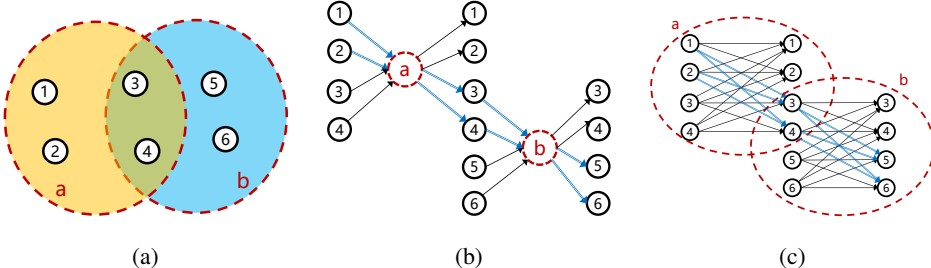

(a)            (b)            (c)

Figure 1: Different message passing paradigms. **(a) Hypergraph**: An example hypergraph comprising two hyperedges $e_a = \{v_1, v_2, v_3, v_4\}, e_b = \{v_3, v_4, v_5, v_6\}$. Messages passing from nodes $v_1, v_2$ to $v_5, v_6$ is along a 2-walk $e_a, e_b$. **(b) Message Passing on Star Expansion**: Contemporary methodologies predominantly treat hyperedges as special nodes and convert any walk via them into a 1-walk (e.g. $\{v_1, v_2, v_a\}, \{v_a, v_3, v_4\}, \{v_3, v_4, v_b\}, \{v_b, v_5, v_6\}$), leading to information squashing when passing messages through the bottled hyperedges (e.g. $v_a, v_b$). **(c) Hypergraph-Native Message Passing**: Our method preserves the high-order structures of the hypergraph during message passing, thereby maintaining the richness of interactions.

order structural benefits of hypergraphs, masks the distinctive contributions of individual nodes, and consequently leads to compromised discriminative representations, especially when dealing with large hyperedges that bridge a heterophilic collection of nodes (Wang et al., 2023a).

To address the aforementioned limitation, we introduce the Hypergraph-native Message Passing (HMP) framework, which is architected to harness the higher-order structure of hypergraphs more efficiently by utilizing full-rank interactions across the incidences (or the hyperedge-node pairs) of the hypergraph and its dual (Huang et al., 2020), as depicted in Figure 1c. This allows nodes to engage directly with their immediate neighbours without the information squashing through hyperedges that might result in the loss of critical information (Yang et al., 2020) and, likewise, resolves the squashing of hyperedges' information through node-level bottlenecks. While being compatible with conventional node- and hyperedge-centric approaches, HMP's incidence-centric perspective is particularly advantageous for incidence-level tasks, such as hyperedge-dependent labelling, where a node can assume different labels depending on the hyperedge it is associated with (Yoon et al., 2020; Choe et al., 2023).

In essence, our contributions are **(1)** a pioneering learning paradigm, HMP, that exploits the high-order structures of hypergraphs for message propagation, **(2)** a novel incidence-centric perspective for incidence-, hyperedge-, and node-level tasks, and **(3)** state-of-the-art performance on 6 hyperedge-dependent labelling tasks and 13 node classification benchmarks.

## 2   PRELIMINARIES

Denoting $[n] = \{1, 2, \ldots, n\}$, we represent a hypergraph as $\mathcal{H} = (\mathcal{V}, \mathcal{E})$, where $\mathcal{V} = \{v_j | j \in [|\mathcal{V}|]\}$ denotes the collection of nodes and $\mathcal{E} = \{e_i | i \in [|\mathcal{E}|]\}$ denotes the collection of hyperedges. Each hyperedge $e_i$ within $\mathcal{E}$ is a subset of $\mathcal{V}$, indicating the nodes that are interconnected by said hyperedge. Given that a hyperedge can connect more than two nodes, the hypergraph is characterized by an incidence matrix $\boldsymbol{B} \in \{0, 1\}^{|\mathcal{E}| \times |\mathcal{V}|}$, where $B_{ij} = 1$ signifies $v_j \in e_i$, and $B_{ij} = 0$ otherwise. Node features are encapsulated in a matrix $\boldsymbol{X} \in \mathbb{R}^{|\mathcal{V}| \times d_v}$, with $d_v$ denoting the dimensionality of the node feature space. The $i$-th row of $\boldsymbol{X}$, denoted as $\boldsymbol{X}_{i,:}$, represents the feature vector associated with node $v_i$. Hyperedge features, when present, are stored in a matrix $\boldsymbol{E} \in \mathbb{R}^{|\mathcal{E}| \times d_e}$, where $d_e$ is the dimensionality of the hyperedge feature space. The $i$-th row of $\boldsymbol{E}$, denoted as $\boldsymbol{E}_{i,:}$, corresponds to the feature vector of hyperedge $e_i$. Incidence attributes are represented by a tensor $\mathbf{B} \in \mathbb{R}^{|\mathcal{E}| \times |\mathcal{V}| \times d_b}$, where $d_b$ specifies the dimensionality of the incidence attribute space. The sub-tensor $\boldsymbol{B}_{i,j,:}$ represents the attribute vector associated with the incidence of hyperedge $e_i$ and node $v_j$.

We introduce the concepts of $s$-walks (Aksoy et al., 2020) and duality (Huang et al., 2020) within the context of hypergraphs to help understand the message passing paradigms.

**Definition 1** (*s*-walk). *For a positive integer $s$, an $s$-walk of length $k$ from hyperedge $e_s$ to $e_t$ in a hypergraph is defined as a sequence of hyperedges $e_{i_0}, e_{i_1}, \ldots, e_{i_k}$, where $i_0 = s, i_k = t$, and $\min_{j \in [k]} |e_{i_{j-1}} \cap e_{i_j}| = s$.*

In this definition, the parameter $s$ governs the extent of hyperedge intersections and can be conceptualized as the 'bandwidth' through which a message traverses along an $s$-walk. It is evident that traditional paths in standard graphs are special cases of $s$-walks, specifically when $s = 1$. For instance, Figure 1a illustrates a 2-walk along $e_a, e_b$ that connects nodes $v_1, v_2$ to $v_5, v_6$. In the standard graph (Figure 1b), a path, such as $v_1, v_a, v_3, v_b, v_5$, is a 1-walk.

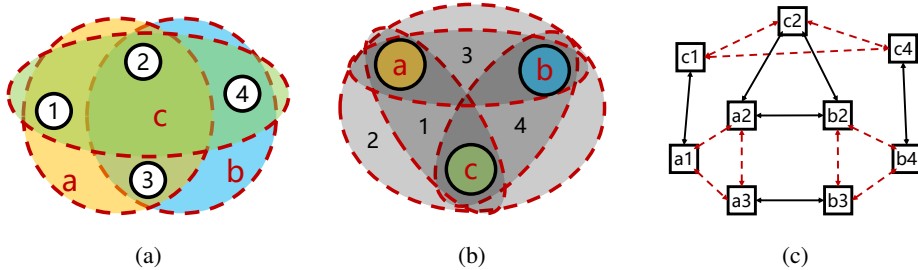

|     |     |     |
| :-: | :-: | :-: |
| (a) | (b) | (c) |

Figure 2: **(a) The Original Hypergraph**. **(b) The Dual Hypergraph**. **(c) Incidence-Centric Message Passing**: HMP facilitates the message passing (MP) on both duals of a hypergraph. Red dashed lines indicate MP along the original hypergraph, and black solid lines represent MP along the dual hypergraph.

**Definition 2** (Dual Hypergraph). *The dual of a hypergraph $\mathcal{H} = (\mathcal{V}, \mathcal{E})$ is denoted as $\mathcal{H}^* = (\mathcal{V}^*, \mathcal{E}^*)$, where the nodes of the dual hypergraph are given by $\mathcal{V}^* = \{v_i^* = e_i | i \in [|\mathcal{E}|]\}$ and the hyperedges are defined as $\mathcal{E}^* = \{e_j^* = \{v_i^* | v_j \in e_i\} | j \in [|\mathcal{V}|]\}$.*

In this duality, if a node $v_j$ is contained within a hyperedge $e_i$ in the original hypergraph $\mathcal{H}$, then the corresponding dual hypergraph $\mathcal{H}^*$ will have the node $v_i^*$ contained within the hyperedge $e_j^*$. For instance, Figure 2a depicts a hypergraph comprising four nodes $v_1, v_2, v_3, v_4$ and three hyperedges $e_a = \{v_1, v_2, v_3\}, e_b = \{v_2, v_3, v_4\}, e_c = \{v_1, v_2, v_4\}$. In the dual hypergraph (Figure 2b), the original hyperedges are transformed into nodes, and the original nodes become hyperedges, resulting in the following mappings: $e_1^* = \{v_a^*, v_c^*\}, e_2^* = \{v_a^*, v_b^*, v_c^*\}, e_3^* = \{v_a^*, v_b^*\}, e_4^* = \{v_b^*, v_c^*\}$.

## 3 METHODOLOGY

### 3.1 HYPERGRAPH-NATIVE MESSAGE PASSING

Our first innovation is the Hypergraph-native Message Passing (HMP) framework that harnesses the high-order structural properties of hypergraphs to facilitate unsquashed information exchange. Unlike traditional message passing (MP) approaches that compute node representations along the 'node-hyperedge-node' path, as depicted in Figure 1b, we employ a multiset-to-multiset model $f$ (referred to as the message exchanger) to compute node representations directly between 'node-node' shortcuts as follows:

$$\{\boldsymbol{H}_{i,j,:} | v_j \in e_i\} = f(\{\boldsymbol{X}_{j,:} | v_j \in e_i\}), \forall e_i \in \mathcal{E} \tag{1}$$

where the resulting outputs $\{\boldsymbol{H}_{i,j,:} | v_j \in e_i\}$ represent hyperedge-dependent node representations and maintain the same order as the input features $\{\boldsymbol{X}_{j,:} | v_j \in e_i\}$.

It is worth noting that the conventional MP on the star expansion (SE), such as AllSet (Chien et al., 2021), is a specific instance of (1) when $f$ is implemented as a multiset function. This implementation squashes any walk of a hypergraph into a 1-walk with its bypassing hyperedges as bottlenecks. For example, with star expansion in Figure 1b, the representation of node $v_5$ (or $v_6$) is dependent on the representation of hyperedge $e_b$ (or $e_a$) with a fixed number $h$ of dimensions, which is incapable of conveying rich information when the hyperedges bridge a massive array of heterophilic

nodes (Aponte et al., 2022; Zheng & Worring, 2024). On the contrary, implementing $f$ as a multiset-to-multiset model as in HMP maintains the $s$-walk structure. In the example of Figure 1c, the representation of node $v_5$ is dependent on the combined representation of $v_3$ and $v_4$ with a total of $h \times s$ times dimensions ($s = 2$), thereby alleviating the hyperedge bottleneck issue.

More formally, we demonstrate that the following enhanced MP on SE is a special case of HMP when learning node representations:

**Theorem 1.** *HMP is expressive enough to represent MP on SE with adaptive representation size for hyperedges, defined as*

$$\boldsymbol{X}'_{j,:} = f_{\mathcal{E} \to \mathcal{V}}(\{\boldsymbol{e}'_i | v_j \in e_i\}), \quad \boldsymbol{e}'_i = f_{\mathcal{V} \to \mathcal{E}}(\{\boldsymbol{X}_{j,:} | v_j \in e_i\}),$$

*where $f_{\mathcal{E} \to \mathcal{V}}, f_{\mathcal{V} \to \mathcal{E}}$ are multiset functions, $\boldsymbol{e}'_i$ is an $(h \times |e_i|)$-dimensional representation for hyperedge $e_i$, and $\boldsymbol{X}' \in \mathbb{R}^{|\mathcal{V}| \times h}$ is the obtained node representations.*

The proof is in Appendix C.1. Thus, HMP alleviates the native structure of hypergraphs to facilitate adaptive message passing within hyperedges of varying scales, resulting in stronger expressiveness rather than the fixed-size MP approaches.

### 3.2 INCIDENCE-CENTRIC LEARNING ON HYPERGRAPH DUALS

Our second innovation is the incidence-centric learning mechanism that leverages the symmetric attributes of hypergraphs for thorough propagation. Different from existing approaches like AllSet, which aggregate hyperedge-dependent representations (1) using multiset functions to derive node-level representations and thus introduce node-level bottlenecks, we enhance HMP by incorporating an incidence-centric learning paradigm, which involves applying an additional message exchanger $f^{(e)}$ to node-dependent hyperedges. The message exchanger $f^{(e)}$ applied to hyperedges associated with node $v_j$, as viewed from the dual hypergraph $\mathcal{H}^*$, is expressed as:

$$\{\boldsymbol{H}^{(e)}_{i,j,:} | v^*_i \in e^*_j\} = f^{(e)}(\{\boldsymbol{E}_{i,:} | v^*_i \in e^*_j\}), \forall e^*_j \in \mathcal{E}^*. \tag{2}$$

The outputs $\mathsf{H}^{(e)} \in \mathbb{R}^{|\mathcal{E}| \times |\mathcal{V}| \times h}$ from (2) are added to the outputs of (1) to form the incidence representations $\mathsf{H} \in \mathbb{R}^{|\mathcal{E}| \times |\mathcal{V}| \times h}$. These incidence representations can be recursively fed into (1) and (2), by replacing $\boldsymbol{E}_{i,:}$ and $\boldsymbol{X}_{j,:}$ with $\boldsymbol{H}_{i,j,:}$, to facilitate the propagation of information between hyperedge-dependent nodes and node-dependent hyperedges, resulting in the incidence-centric message passing on hypergraph duals. We formally describe this algorithm in Appendix A and illustrate it in Figure 2c.

We instantiate the multiset-to-multiset message exchangers in HMP as self-attention modules (Vaswani et al., 2017) due to their widely validated effectiveness across multiple domains of deep learning (Kalyan et al., 2021; Han et al., 2020), as

$$f(\boldsymbol{X}) = \text{softmax}(\frac{\boldsymbol{X}\boldsymbol{W}_q \cdot (\boldsymbol{X}\boldsymbol{W}_k)^T}{\sqrt{h}}) \cdot \boldsymbol{X}\boldsymbol{W}_v, \tag{3}$$

where $\boldsymbol{W}_q, \boldsymbol{W}_k, \boldsymbol{W}_v$ are trainable parameters. The self-attention modules help facilitate full-rank interactions within incidences, but with a higher complexity:

**Theorem 2.** *The computation complexity of HMP with (3) is $\max_{e \in \mathcal{E} \cup \mathcal{E}^*} |e|$ times that of AllSet.*

The proof is in Appendix C.2. Fortunately, self-attention has undergone a series of efficient improvements (Tay et al., 2023; Choromanski et al., 2021) that can be applied to HMP. Specifically, we implement the attention modules with linear computation complexity (Tay et al., 2023) (i.e. Performer (Choromanski et al., 2021)) when the message exchanging contexts (hyperedges or nodes) are large, reducing the complexity of HMP to the same as conventional HNNs like AllSet. Moreover, we parallelize the attention module in different hyperedges (or nodes) via the neighbourhood partitioning technique (Luo et al., 2025) to further enhance HMP's memory- and time-efficiencies.

### 3.3 VIRTUAL INCIDENCES FOR HYPEREDGE- AND NODE-LEVEL TASKS

Our third innovation involves virtual incidences that adapt the incidence-centric HMP to hyperedge- and node-level downstream tasks. Different from existing methods that aggregate incidence representations with multiset functions to obtain hyperedge- and node-level representations, we augment

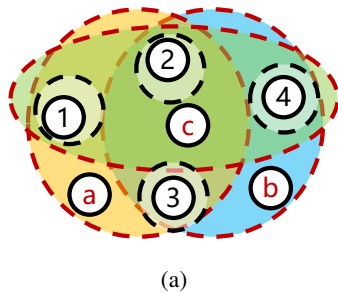 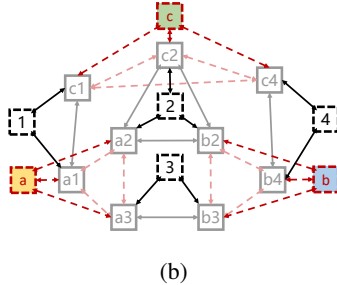

|     |     |
| --- | --- |
| (a) | (b) |

Figure 3: **(a) Extending Hypergraphs (e.g. Figure 2a) With Virtual Nodes and Hyperedges**: A black dashed circle represents a virtual hyperedge encompassing a singular node. A black solid circle with an enclosed red alphabet denotes a virtual node, with the alphabet signifying the hyperedge to which it belongs. **(b) Extended Interactions Involving Virtual Incidences**: The representations of the introduced virtual incidences within HMP preserve both hyperedge- and node-level information, which can be leveraged for subsequent downstream tasks.

the incidence matrix $\boldsymbol{B}$ to $\boldsymbol{B}^+ = \begin{bmatrix} \boldsymbol{B} & \boldsymbol{I}_{|\mathcal{E}|} \\ \boldsymbol{I}_{|\mathcal{V}|} & \boldsymbol{0} \end{bmatrix}$, where $\boldsymbol{I}_n$ denotes an $n \times n$ identity matrix, to make HMP compatible with hyperedge- and node-level tasks. As exemplified in Figure 3, the presence of $\boldsymbol{I}_{|\mathcal{E}|}$ in the first row of $\boldsymbol{B}^+$ signifies the addition of a virtual node corresponding to each hyperedge, which we refer to as virtual hyperedge-incidence. Furthermore, the $\boldsymbol{I}_{|\mathcal{V}|}$ in the first column indicates the insertion of a virtual hyperedge for each node, termed virtual node-incidence. Likewise, we augment the incidence attributes $\mathsf{B}$ to $\mathsf{B}^+ \in \mathbb{R}^{(|\mathcal{E}|+|\mathcal{V}|) \times (|\mathcal{V}|+|\mathcal{E}|) \times d_b}$ by adding distinguishable attributes of virtual incidences. Our strategies of augmenting $B$ and $\mathsf{B}$ are flexible for specific downstream tasks, as detailed in Appendix B.

The advantage of augmenting hypergraphs with virtual incidences is multifaceted: not only does it enable the utilization of representations of virtual hyperedge-incidences for hyperedge-level tasks and those of virtual node-incidences for node-level tasks, but it also implicitly provides hyperedge- and node-level information during the incidence-level learning process of HMP. These implicit informations have been demonstrated to be advantageous in our empirical investigations in Appendix E. Besides, with the introduction of virtual hyperedge- and node-incidences, HMP can emulate the functionality of AllSet (Chien et al., 2021), as demonstrated by the following theorem:

**Theorem 3.** *AllSet is a special case of HMP with virtual incidences.*

The proof is in Appendix C.3. With the known fact that AllSet encompasses the propagation rules of numerous existing Hypergraph Neural Networks (Chien et al., 2021), including HyperGCN (Yadati et al., 2019), HGNN (Feng et al., 2019), HCHA (Bai et al., 2021), HNHN (Dong et al., 2020), and HyperSAGE (Arya et al., 2020), HMP is a more generalized case of them.

## 4 RELATED WORKS

Among the proposed Hypergraph Neural Networks (HNNs) (Kim et al., 2024), many approaches (Dong et al., 2020; Huang & Yang, 2021; Georgiev et al., 2022) predominantly employ the star expansion technique, where hyperedges are treated as special nodes as depicted in Figure 1b, effectively transforming hypergraphs into conventional graphs to apply traditional Graph Neural Networks (GNNs) that operate on a node-wise message passing paradigm. For instance, AllSet (Chien et al., 2021) extends the message passing framework to hypergraphs and encompasses the propagation rules of most existing HNNs, including HyperGCN (Yadati et al., 2019), HGNN (Feng et al., 2019), HCHA (Bai et al., 2021), HNHN (Dong et al., 2020), and HyperSAGE (Arya et al., 2020). AllSet constructs its layers using two multiset functions: one aggregates node messages to form hyperedge representations, and the other aggregates hyperedge messages to update node representations. This leads to the derivation of AllDeepSets and AllSetTransformer when the multiset functions are instantiated as DeepSets (Zaheer et al., 2017) and SetTransformer (Lee et al., 2019), respectively. In AllSet, a node can only indirectly obtain information from its direct neighbours

through the 'special hyperedge node', leading to information bottlenecks. This is because it squashes diverse messages from multiple nodes into a single hyperedge representation, a phenomenon known as 'over-squashing' (Di Giovanni et al., 2023). In contrast, we propose the Hypergraph-native Message Passing (HMP) framework to enable direct message exchanging among nodes within the same hyperedge and overcome the hyperedge bottleneck issue, as illustrated in Figure 1c. This allows for a more efficient and effective information flow within the hypergraph structure, potentially leading to better performance on various hypergraph-based tasks.

Recently, a growing number of hypergraph methods have recognized, to varying degrees, the importance of the incidence-centric learning paradigm. WHATsNET (Choe et al., 2023) applies self-attention mechanisms to nodes within the same hyperedge, sharing conceptual similarities with HMP. Nevertheless, following the application of self-attention, WHATsNET continues to aggregate node information to form hyperedge representations, aligning with the AllSet framework and thus encountering the same hyperedge bottlenecks as AllSet does. While HyperGT (Liu et al., 2023b) also recognizes the value of incidence-level interactions, arguing they outperform conventional node-hyperedge-node message passing, its approach transforms the hypergraph into a graph of original nodes and 'hyperedge nodes' with an adjacency matrix as $\begin{bmatrix} \mathbf{0} & \mathbf{B} \\ \mathbf{B}^T & \mathbf{0} \end{bmatrix}$, where $\mathbf{B}$ is the incidence matrix of the original hypergraph. Then HyperGT applies GAT (Velickovic et al., 2018) on this expanded graph, thus maintaining a node- and hyperedge-centric method. Yang et al. (2020) has criticized the star expansion performed at the hyperedge-/node-level for its information loss and has proposed an alternative, the line expansion (LE). LE treats each node-hyperedge pair as an individual node and creates connections between these nodes, once any two of them share either nodes or hyperedges in the original graph. Then LE applies traditional GNNs to the resulting graphs to develop its implementations, such as the development of LEGCN by applying GCN (Kipf & Welling, 2016). While LE's node-hyperedge pairs are conceptually akin to our paradigm, LE treats the converted graph as homogeneous, which leads to confusion on connections between incidences that share nodes versus those that share hyperedges. This limitation is resolved by CoNHD (Zheng & Worring, 2024), which inherently distinguishes between LE's two types of relationships and processes them separately, maintaining a more nuanced understanding of the hypergraph structure. However, CoNHD, describing its message passing framework with the diffusion concept, implements its weighted version of diffusion operators as SetTransformers with a fixed number $k$ of inducing points. This is equivalent to enlarging the dimensions of hyperedge representations $k$ times in message passing on star expansion, and thus cannot have adaptive dimensions for hyperedges with different scales. In contrast, HMP utilizes a switchable attention module (Luo et al., 2025) that reserves the adaptiveness and is a true incidence-centric method.

## 5 EXPERIMENTS

We present three experimental validations on the effectiveness of HMP for **(1)** heterophilic $s$-walks, **(2)** incidence-level tasks (i.e. hyperedge-dependent labelling), and **(3)** node-level tasks (i.e. node classification). Other studies on HMP, including ablation studies on virtual incidences, hyperparameter sensitivity analysis, and the tuning strategy, are in Appendix E and Appendix F.

### 5.1 ON THE SYNTHETIC HYPERCHAIN DATASETS

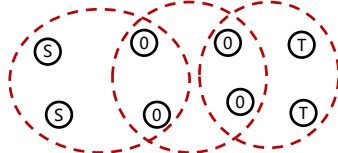 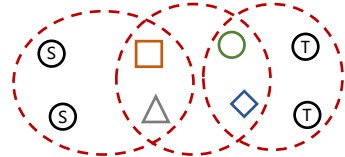

Figure 4: The synthesized 2-walk hyperchains of length 3. Label information is encoded at the source nodes (denoted as 'S') using one-hot vectors. The objective is to propagate and recover the identical labels at the target nodes (denoted as 'T'), which are initialized with all-zero feature vectors. The intermediate nodes along the 2-walks are **(left)** initialized with all-zero features in the homophilic setting, or **(right)** assigned randomized one-hot vectors in the heterophilic setting.

In this section, we construct the Hyperchains datasets to highlight the benefits of HMP over traditional Message Passing (MP) methods. As depicted in Figure 4, a hyperchain is structured with $k$ hyperedges, each comprising $2s$ nodes. These hyperedges are sequentially overlapped, with each consecutive pair sharing $s$ nodes. Consequently, the hyperchain is structured to represent an $s$-walk of length $k$. Within a hyperchain, the initial $s$ nodes (referred to as sources) and the final $s$ nodes (referred to as targets) are assigned the same label. This label is represented as one-hot vectors at the source nodes. The objective is to predict the labels at the target nodes, which are initialized with all-zero feature vectors. By initializing the intermediate nodes along the $s$-walks with either all-zero features or randomized one-hot vectors, we can simulate both homophilic and heterophilic settings. In the homophilic setting, nodes within the same hyperedge are similar, while in the heterophilic setting, they are dissimilar. This setup allows us to evaluate the performance of HMP and other MP methods under different homophily structures.

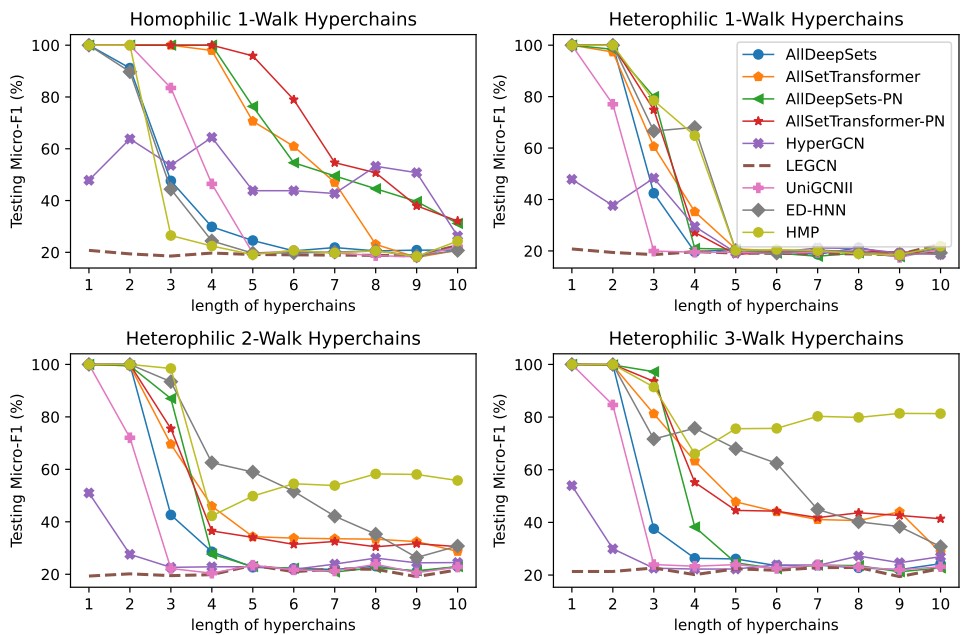

Figure 5: Averaged scores across 10 runs on the Hyperchains datasets with different widths, lengths, and homophily settings.

We benchmark HMP against several baselines, including AllDeepSets, AllSetTransformer (Chien et al., 2021), HyperGCN (Yadati et al., 2019), LEGCN (Yang et al., 2020), UniGCNII (Huang & Yang, 2021), and ED-HNN (Wang et al., 2023a). For AllDeepSets and AllSetTransformer, we introduce a PairNorm-like (Zhao & Akoglu, 2020) normalization technique from Xu et al. (2022) to tackle the over-smoothing issue and derive AllDeepSets-PN and AllSetTransformer-PN. All models have hidden dimensions set to 64 and the number of attention heads (if exists) set to 1. For each setup with varying width $s$, length $k$, and degree of homophily, we construct 1,000 hyperchains, which are then distributed into a training set of 500, a validation set of 250, and a testing set of 250. Each method is configured with a specific number of layers to precisely convolute information from source nodes to target nodes. They are trained, with NVIDIA GeForce RTX 3060 (12GB), for 1,000 epochs on the training set, and the accuracy on the testing set is recorded as the performance metric for the run if it corresponds to the best validation set performance. The average scores from 10 runs are presented in Figure 5. The results in the first subfigure reveal that HMP is not the most effective for message passing on homophilic hyperchains. Indeed, both AllSetTransformer and UniGCNII outperform HMP on nearly all homophilic hyperchains, regardless of the hyperchain length. HyperGCN also exhibits robust performance as the hyperchains become longer. Besides, PairNorm (PN) shows significant improvements on AllDeepSets and AllSetTransformer, demonstrating its effectiveness in long-range propagations with homophilic settings. However, in the case of heterophilic hyperchains, all methods, including AllSet with PairNorm, experience a significant decline in performance as the length increases. Among the methods tested, ED-HNN and HMP prove to be the

most resilient, highlighting their advantage in learning under heterophily. Furthermore, as our analysis indicates, conventional methods are unable to fully utilize the structural benefits of hypergraphs as the hyperchain width $s$ increases, due to the hyperedge bottlenecks. In contrast, as shown in the third and fourth subfigures, HMP demonstrates improved long-range propagation with increasing width $s$ and is the only method that maintains robustness as the hyperchain lengthens. In summary, HMP exhibits a more potent learning capability in heterophilic contexts compared to traditional baselines and is uniquely positioned to harness the structural potential of hypergraphs.

## 5.2 HYPEREDGE-DEPENDENT LABELLING

Table 1: Averaged Micro-F1 (the upper section) and Macro-F1 (the lower section) scores across five runs for hyperedge-dependent labelling. The best score for each dataset is **bolded**, the second best is underlined, and the third is *italic*. The '8 Baselines' are HNHN, HGNN, HCHA, HAT, UniGCNII, HNN, HST, and AllSetTransformer, with their detailed performance and the datasets' characteristics summarized in Appendix D.

| | Coauth DBLP | Coauth AMiner | Email Enron | Email Eu | Stack Biology | Stack Physics |
|---|---|---|---|---|---|---|
| 8 Baselines | 56.4±0.4 | 59.6±0.7 | 81.7±0.1 | 67.1±0.1 | 69.4±0.2 | 75.5±1.0 |
| WHATsNET | 60.5±0.2 | 63.0±0.5 | *82.6±0.1* | 67.1±0.0 | 74.2±0.3 | 77.0±0.3 |
| CoNHD | *62.0±0.2* | *65.0±0.3* | 91.1±0.1 | *70.9±0.1* | *74.9±0.2* | *77.7±0.1* |
| **HMP** (100 epochs) | 77.6±0.6 | 72.5±0.2 | 72.4±0.3 | 71.4±0.9 | 77.2±0.2 | 77.9±0.7 |
| **HMP** | **89.4±0.2** | **80.4±0.1** | **93.4±1.51** | **78.2±0.1** | **78.8±0.2** | **82.6±0.2** |
| 8 Baselines | 54.9±0.3 | 58.3±0.8 | 75.3±0.4 | 64.0±0.2 | 63.1±0.6 | 66.6±1.3 |
| WHATsNET | 59.5±0.2 | 62.3±0.7 | *76.0±0.4* | 64.6±0.3 | 68.6±0.4 | 70.7±0.4 |
| CoNHD | *60.4±0.2* | *64.6±0.4* | 87.1±0.2 | *69.0±0.2* | *69.5±0.4* | *71.2±0.5* |
| **HMP** (100 epochs) | 77.3±0.6 | 71.7±0.3 | 67.1±0.3 | 71.0±0.7 | 73.0±0.3 | 73.7±1.0 |
| **HMP** | **89.3±0.2** | **79.9±0.1** | **91.3±2.1** | **77.8±0.1** | **76.0±2.0** | **79.1±0.2** |

In this section, we evaluate the effectiveness of HMP on the hyperedge-dependent labelling tasks, which aim to assign different labels to nodes based on the hyperedges to which they belong. The datasets include Coauth-DBLP, Coauth-AMiner, Email-Enron, Email-Eu, Stack-Biology, and Stack-Physics (Choe et al., 2023). In Table 1, we present the Micro- and Macro-F1 scores for HMP, trained with NVIDIA A800 (80GB), and a range of baseline methods, including HNHN (Dong et al., 2020), HGNN (Feng et al., 2019), HCHA (Bai et al., 2021), HAT (Hwang et al., 2021), UniGCNII (Huang & Yang, 2021), HNN (Aponte et al., 2022), HypergraphSetTransformer (HST) (Choe et al., 2023), AllSetTransformer (Chien et al., 2021), WHATsNET (Choe et al., 2023), and CoNHD (Zheng & Worring, 2024). The experimental setup was consistent with that of Choe et al. (2023), with the exception that HMP was trained in full-batch for a maximum of 10,000 epochs with a patience of 1,000 epochs for early stopping, whereas the baselines were trained in mini-batch for 100 epochs. This difference in training epochs does not compromise the fairness of the comparison, because the baselines have sufficient gradient descent steps for their optimization. For instance, on the Coauth-AMiner dataset with 1,712,433 nodes, the baselines were trained using node batches of size 512, allowing for 334,500 gradient descent updates in 100 epochs. Nonetheless, the results of HMP with only 100 training epochs are also included in the table for reference.

As indicated in Table 1, HMP achieves significant improvements over the current state-of-the-art methods, WHATsNET and CoNHD, in terms of both Micro-F1 and Macro-F1 scores across various datasets. The margins range from 2.3% of Micro-F1 on Email-Enron to an impressive 28.9% of Macro-F1 on Coauth-DBLP. Even with insufficient training of just 100 epochs, HMP still manages to outperform the baselines on 5 out of the 6 tested benchmarks, with underperformance only on the Email-Enron dataset. This clearly illustrates the efficacy of the incidence-centric HMP for hyperedge-dependent labelling tasks.

## 5.3 NODE CLASSIFICATION

In this section, we extend the application of HMP to node classification tasks by leveraging virtual incidences, and we benchmark it against the state-of-the-art (SotA) hypergraph neural networks

(HNNs). We have gathered a comprehensive set of baselines, including Multi-Layered Perceptron (MLP), HGNN (Feng et al., 2019), UniGCNII (Huang & Yang, 2021), HAN (Wang et al., 2019) (trained in full-batch and mini-batch), HyperGT (Liu et al., 2023b), AllSet (Chien et al., 2021) (including AllDeepSets and AllSetTransformer), ED-HNN (Wang et al., 2023a) (with its ED-HNNII variant), and PhenomNN (Wang et al., 2023b) (with its PhenomNN$_{simple}$ variant). We use 13 datasets: Cora, Citeseer, Pubmed, Cora-CA, DBLP-CA (Yadati et al., 2019), 20Newsgroups, ModelNet40 (Wu et al., 2014), NTU2012 (Chen et al., 2003), Yelp, House (Chodrow et al., 2021), Walmart (Amburg et al., 2019), Senate, and Congress (Wang et al., 2023a). These datasets vary in numerous ways, with a particular focus on the degree of homophily, which measures the probability of connected nodes sharing the same labels. We calculate the homophily score (Pei et al., 2020) based on the clique expansion (CE) of the hypergraphs. A higher CE homophily score indicates that two connected nodes in the hypergraph are more likely to be similar, reflecting a higher degree of homophily within the network.

Table 2: Averaged and standard deviation of Micro-F1 scores (%) across ten runs for node classification. Scores of HMP are **bolded** if HMP surpasses the best SotA (from Chien et al. (2021); Wang et al. (2023a;b); Liu et al. (2023b)), and are underlined if HMP surpasses the second best. More characteristics of the datasets and detailed performance of the baselines are in Appendix D.

| Dataset | CE Homo. | State of the Art (SotA) | | | | **HMP** |
| | | Rank-1 | | Rank-2 | | |
| | | Method | Micro-F1 | Method | Micro-F1 | |
| --- | --- | --- | --- | --- | --- | --- |
| Pubmed | 0.952 | ED-HNN | 89.56±0.62 | AllSet | 88.75±0.33 | 88.45±0.38 |
| Cora | 0.897 | PhenomNN | 82.29±1.42 | ED-HNN | 80.31±1.35 | 80.35±1.32 |
| Citeseer | 0.893 | PhenomNN | 75.10±1.59 | HAN | 74.12±1.52 | 73.82±1.21 |
| DBLP-CA | 0.869 | ED-HNN | 91.93±0.29 | PhenomNN | 91.91±0.21 | 91.87±0.20 |
| ModelNet40 | 0.853 | PhenomNN | 98.66±0.20 | AllSet | 98.20±0.20 | 98.54±0.26 |
| Cora-CA | 0.803 | PhenomNN | 85.81±0.90 | HAN | 84.04±1.02 | 84.61±1.35 |
| NTU2012 | 0.752 | PhenomNN | 91.03±1.04 | UniGCNII | 89.30±1.33 | 90.70±1.06 |
| Congress | 0.555 | HyperGT | 95.23±0.73 | ED-HNN | 95.19±1.34 | **95.77±1.15** |
| Walmart | 0.530 | HyperGT | 69.83±0.39 | ED-HNN | 67.24±0.45 | **72.42±0.46** |
| House | 0.509 | HyperGT | 74.55±1.99 | ED-HNN | 73.95±1.97 | 74.09±1.95 |
| Senate | 0.498 | HyperGT | 65.49±5.11 | ED-HNN | 64.79±5.14 | **68.31±5.32** |
| 20Newsgroups | 0.461 | PhenomNN | 81.74±0.52 | MLP | 81.42±0.49 | 81.64±0.44 |
| Yelp | 0.226 | AllSet | 36.89±0.51 | HGNN | 33.04±0.62 | 36.48±0.44 |

With the experimental settings identical to Wang et al. (2023a), we evaluate the Micro-F1 scores of HMP for the node classification tasks and report them in Table 2. As can be seen in the table, while HMP with virtual incidences lags behind the best SotA methods, namely ED-HNN and PhenomNN, with noticeable margins on the three most homophilic datasets, Cora, Citeseer, and Pubmed, it significantly outperforms SotA HNNs on Congress, Walmart, and Senate, and surpasses the second-best baselines on other datasets as the CE homophily scores decrease. This is because the propagation rules of ED-HNN and PhenomNN are diffusion-based, with the inductive bias effective for homophilic graphs. On the contrary, HMP adapts to wider scenarios without the limitation of such homophilic assumptions, instead with competitive advantages in tackling node-level tasks characterized by heterophily.

## 6 CONCLUSIONS

We introduce Hypergraph-native Message Passing (HMP), a learning paradigm that harnesses the high-order structural properties of hypergraphs to effectively alleviate information bottlenecks and facilitate information propagation. Its incidence-centric perspective and the augmented virtual incidences position HMP as a competitive solution for incidence-, hyperedge-, and node-level representation learning tasks, demonstrating remarkable efficacy across 6 hyperedge-dependent labelling benchmarks and 13 node classification datasets.

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

## A   THE ALGORITHM OF HMP

We delineate the incidence-centric algorithm of Hypergraph-native Message Passing (HMP) in Algorithm 1. In line 1, we initialize the incidence representations based on the incidence attributes. Lines 2 through 4 entail the application of a message exchanger $f_0^{(v)}$ to hyperedge-dependent nodes, which converts node features into incidence representations. Lines 5 to 7 involve the application of an additional message exchanger to node-dependent hyperedges, aimed at transforming hyperedge

---

**Algorithm 1** Hypergraph-Native Message Passing

---

**Input:** the hypergraph $\mathcal{H} = (\mathcal{V}, \mathcal{E})$, the number of layers $l$, node features $\boldsymbol{X}$, hyperedge features
   $\boldsymbol{E}$, incidence attributes **B**, message exchangers $f_0^{(v)}, f_0^{(e)}, f_1^{(v)}, f_1^{(e)}, \ldots, f_l^{(v)}, f_l^{(e)}$.

**Output:** incidence representations **H**.

1: **H** = **B**
2: **for parallel** $i = 1, 2, \ldots, |\mathcal{E}|$ **do**
3:    $\{\boldsymbol{H}_{i,j,:}|v_j \in e_i\}{+}{=}f_0^{(v)}(\{\boldsymbol{X}_{j,:}|v_j \in e_i\})$
4: **end for**
5: **for parallel** $j = 1, 2, \ldots, |\mathcal{V}|$ **do**
6:    $\{\boldsymbol{H}_{i,j,:}|v_i^* \in e_j^*\}{+}{=}f_0^{(e)}(\{\boldsymbol{E}_{i,:}|v_i^* \in e_j^*\})$
7: **end for**
8: **for** $k = 1, 2, \ldots, l$ **do**
9:    $\mathbf{H}^{(v)} = 0$
10:   **for parallel** $i = 1, 2, \ldots, |\mathcal{E}|$ **do**
11:      $\{\boldsymbol{H}_{i,j,:}^{(v)}|v_j \in e_i\} = f_k^{(v)}(\{\boldsymbol{H}_{i,j,:}|v_j \in e_i\})$
12:   **end for**
13:   $\mathbf{H}^{(e)} = 0$
14:   **for parallel** $j = 1, 2, \ldots, |\mathcal{V}|$ **do**
15:      $\{\boldsymbol{H}_{i,j,:}^{(e)}|v_i^* \in e_j^*\} = f_k^{(e)}(\{\boldsymbol{H}_{i,j,:}|v_i^* \in e_j^*\})$
16:   **end for**
17:   $\mathbf{H} {+}{=} \mathbf{H}^{(v)} + \mathbf{H}^{(e)}$
18: **end for**
19: **return H**

---

features into incidence representations. From lines 8 to 18, we alternately apply message exchangers to hyperedge-dependent nodes and node-dependent hyperedges, thereby enabling information propagation across the hypergraph duals. This approach circumvents the node and hyperedge bottlenecks, ensuring a more effective and efficient traversal of the hypergraph's complex interactions.

## B    DETAILS OF VIRTUAL INCIDENCES

While HMP can augment the incidence matrix $\boldsymbol{B}$ to $\boldsymbol{B}^+ = \begin{bmatrix} \boldsymbol{B} & \boldsymbol{I}_{|\mathcal{E}|} \\ \boldsymbol{I}_{|\mathcal{V}|} & \boldsymbol{0} \end{bmatrix}$ for downstream tasks, it has the flexibility to augment the incidence matrix with only virtual hyperedge-incidences or virtual node-incidences, rather than both, depending on the requirements of the tasks. Furthermore, the augmentation of the incidence attributes **B** to $\mathbf{B}^+ \in \mathbb{R}^{(|\mathcal{E}|+|\mathcal{V}|) \times (|\mathcal{V}|+|\mathcal{E}|) \times d_b}$ can also be flexible, provided that it effectively differentiates among the original attributes and the added different types of virtual incidences. For instance, in the case (e.g. our node classification tasks) where incidence attributes **B** = 0, we assign fixed 2-dimensional encodings $(1, 0)$ to virtual hyperedge-incidences, $(0, 1)$ to virtual node-incidences, and $(0, 0)$ to original incidences to construct $\mathbf{B}^+$. When incidence attributes exist (**B** $\neq$ 0), such as in our hyperedge-dependent labelling tasks, we concatenate the aforementioned 2-dimensional encodings to $\begin{bmatrix} \mathbf{B} & 0 \\ 0 & 0 \end{bmatrix}$ to augment the original incidence attributes. These encodings are fixed but will be transformed into learned embeddings after an encoding layer.

## C    THEORETICAL ANALYSIS

### C.1    THE PROOF OF THEOREM 1

*Proof.* The idea is to store hyperedge representations of MP on SE into the involved incidence representations of HMP. Assume that $\phi$ is a rearranging function which receives an $(h \times s)$-dimensional vector, with $s$ being a variable, and outputs an ordered multiset of $s$ elements, where each element is an exclusive $h$-dimensional slice of the input. We construct a multiset-to-multiset model $f$ in HMP

as the composite function of $\phi \circ f_{\mathcal{V} \to \mathcal{E}}$ and get hyperedge-dependent node representations as

$$\{\boldsymbol{H}_{i,j,:}|v_j \in e_i\} = \phi(f_{\mathcal{V} \to \mathcal{E}}(\{\boldsymbol{X}_{j,:}|v_j \in e_i\})) = \phi(\boldsymbol{e}'_i), \forall e_i \in \mathcal{E}.$$

Aggregating them by nodes with the multiset function: $v_j \to f_{\mathcal{E} \to \mathcal{V}}(\phi^{-1}(\{\boldsymbol{H}_{i,k,:}|v_k \in e_i\})|v_j \in e_i), \forall v_j \in \mathcal{V}$ obtains the same node representations $\boldsymbol{X}'$ produced by the previously defined MP on SE. $\square$

## C.2 THE PROOF OF THEOREM 2

*Proof.* Since HMP facilitates full message exchanging among incidences within the same hyper-edges and nodes, with self-attention's quadratic computation complexity, it has a total complexity proportional to

$$\sum_{e \in \mathcal{E}} |e|^2 + \sum_{e^* \in \mathcal{E}^*} |e^*|^2$$

$$\leq \sum_{e \in \mathcal{E}} |e| \cdot \max_{e' \in \mathcal{E}} |e'| + \sum_{e^* \in \mathcal{E}^*} |e^*| \cdot \max_{e' \in \mathcal{E}^*} |e'|$$

$$\leq 2b \cdot \max_{e \in \mathcal{E} \cup \mathcal{E}^*} |e|,$$

where $b = \sum_{e \in \mathcal{E}} |e| = \sum_{e^* \in \mathcal{E}^*} |e^*|$ is the number of incidences.

Since AllSet only passes messages between nodes and their belonging hyperedges, its complexity is $O(b)$. Thus, the complexity of HMP self-attention is $\max_{e \in \mathcal{E} \cup \mathcal{E}^*} |e|$ times that of AllSet. $\square$

## C.3 THE PROOF OF THEOREM 3

*Proof.* Define the update rule of AllSet as

$$\boldsymbol{E}'_{i,:} = f_{\mathcal{V} \to \mathcal{E}}(\{\boldsymbol{X}_{j,:}|v_j \in e_i\}, \boldsymbol{E}_{i,:}), \forall i \in [|\mathcal{E}|],$$

$$\boldsymbol{X}'_{j,:} = f_{\mathcal{E} \to \mathcal{V}}(\{\boldsymbol{E}'_{i,:}|v_j \in e_i\}, \boldsymbol{X}_{j,:}), \forall j \in [|\mathcal{V}|],$$

where $f_{\mathcal{E} \to \mathcal{V}}, f_{\mathcal{V} \to \mathcal{E}}$ are multiset functions with respect to their first inputs and $\boldsymbol{E}', \boldsymbol{X}'$ are representations for hyperedges and nodes. We extend the hypergraph by creating virtual hyper-edges $\{e^+_{|\mathcal{E}|+1}, e^+_{|\mathcal{E}|+2}, \ldots, e^+_{|\mathcal{E}|+|\mathcal{V}|}\}$, where $e^+_{|\mathcal{E}|+j} = \{v_j\}, \forall j \in [|\mathcal{V}|]$, and inserting virtual nodes $\{v_{|\mathcal{V}|+1}, v_{|\mathcal{V}|+2}, \ldots, v_{|\mathcal{V}|+|\mathcal{E}|}\}$ into existing hyperedges to get $e^+_i = e_i \cup \{v_{|\mathcal{V}|+i}\}, \forall i \in [|\mathcal{E}|]$, where each virtual node $v_{|\mathcal{V}|+i}$ is attached with the hyperedge features $\boldsymbol{E}_{i,:}$.

Then we construct the first message exchanger $f_0^{(v)}$ as

$$\{\boldsymbol{H}_{i,j,:}|v_j \in e^+_i\} = f_0^{(v)}(\{\boldsymbol{X}_{j,:}|v_j \in e^+_i\}), \forall i \in [|\mathcal{E}| + |\mathcal{V}|],$$

satisfying that $\boldsymbol{H}_{i,j,:}$ equals to

$$\begin{cases} f_{\mathcal{V} \to \mathcal{E}}(\{\boldsymbol{X}_{k,:}|v_k \in e_i\}, \boldsymbol{E}_{i,:}), & i \in [|\mathcal{E}|], v_j \in e^+_i, \\ \boldsymbol{X}_{j,:}, & j \in [|\mathcal{V}|], i = |\mathcal{E}| + j, \\ 0, & \text{otherwise.} \end{cases}$$

With the intermediate $f_0^{(e)}$ and $f_1^{(v)}$ omitted, we construct the last message exchanger $f_1^{(e)}$ as

$$\{\boldsymbol{H}'_{i,j,:}|v_j \in e^+_i\} = f_1^{(e)}(\{\boldsymbol{H}_{i,j,:}|v_j \in e^+_i\}), \forall j \in [|\mathcal{E}| + |\mathcal{V}|]$$

satisfying that $\boldsymbol{H}'_{i,j,:}$ equals to

$$\begin{cases} f_{\mathcal{E} \to \mathcal{V}}(\{\boldsymbol{H}_{k,j,:}|v_j \in e_k\}, \boldsymbol{H}_{i,j,:}), & j \in [|\mathcal{V}|], i = |\mathcal{E}| + j, \\ \boldsymbol{H}_{i,j,:}, & i \in [|\mathcal{E}|], j = |\mathcal{V}| + i, \\ 0, & \text{otherwise.} \end{cases}$$

Table 3: Datasets for the hyperedge-dependent labelling tasks (the upper part) and the averaged Micro-F1 (the middle part) and Macro-F1 (the lower part) scores of the 8 baselines (Choe et al., 2023). The best score for each dataset is **bolded**.

| Dataset | Coauth DBLP | Coauth AMiner | Email Enron | Email Eu | Stack Biology | Stack Physics |
|---|---|---|---|---|---|---|
| #nodes | 108,484 | 1,712,433 | 21,251 | 986 | 15,490 | 80,936 |
| #hyperedges | 91,266 | 2,037,605 | 101,124 | 209,508 | 26,823 | 200,811 |
| #incidences | 321,011 | 5,129,998 | 1,186,521 | 541,842 | 56,257 | 479,809 |
| #classes | 3 | 3 | 3 | 2 | 3 | 3 |
| HNHN | 48.6±0.4 | 52.0±0.2 | 73.8±2.8 | 64.3±0.4 | 64.0±0.5 | 50.6±5.3 |
| HGNN | 54.0±0.4 | 56.6±0.2 | 72.5±0.4 | 63.3±0.1 | 68.9±0.2 | 68.6±0.4 |
| HCHA | 45.1±0.7 | 46.8±2.0 | 66.6±1.0 | 62.0±0.0 | 58.9±0.7 | 62.2±0.3 |
| HAT | 50.3±0.4 | 54.3±0.2 | **81.7±0.1** | 66.9±0.1 | 66.1±0.5 | 70.8±0.5 |
| UniGCNII | 49.7±0.3 | 52.0±0.1 | 73.4±1.0 | 63.0±0.5 | 61.0±0.4 | 67.1±2.2 |
| HNN | 48.8±0.6 | 54.3±0.2 | 76.3±0.3 | — | 61.8±1.5 | 68.3±0.5 |
| HST | **56.4±0.4** | **59.6±0.7** | 77.9±6.7 | **67.1±0.1** | **69.4±0.2** | **75.5±1.0** |
| AllSetTransformer | 49.5±3.8 | 57.7±0.5 | 79.6±1.4 | 66.6±0.5 | 57.1±5.4 | 72.8±3.9 |
| HNHN | 47.8±0.8 | 51.4±0.2 | 63.7±2.3 | 55.2±1.4 | 59.2±0.6 | 42.2±4.3 |
| HGNN | 51.9±0.2 | 55.1±0.4 | 67.4±0.3 | 53.3±0.8 | 62.4±0.7 | 63.0±0.2 |
| HCHA | 33.4±4.8 | 44.7±4.0 | 46.4±0.2 | 49.7±0.1 | 46.5±6.0 | 48.1±0.7 |
| HAT | 48.3±0.6 | 53.3±0.3 | **75.3±0.4** | 63.8±0.2 | 60.6±0.5 | 64.3±0.9 |
| UniGCNII | 47.6±0.2 | 50.7±0.1 | 65.6±1.0 | 56.5±1.3 | 43.3±0.7 | 49.2±1.6 |
| HNN | 48.2±0.6 | 53.3±0.2 | 67.9±0.7 | — | 56.8±1.3 | 61.7±0.5 |
| HST | **54.9±0.3** | **58.3±0.8** | 68.1±12.3 | **64.0±0.2** | **63.1±0.6** | **66.6±1.3** |
| AllSetTransformer | 48.7±4.0 | 57.0±0.2 | 71.9±2.0 | 62.4±2.1 | 44.6±8.1 | 64.6±4.6 |

Thus, representations of virtual hyperedge-incidences are

$$\boldsymbol{H}'_{i,|\mathcal{V}|+i,:} = \boldsymbol{H}_{i,|\mathcal{V}|+i,:}$$
$$= f_{\mathcal{V}\to\mathcal{E}}(\{\boldsymbol{X}_{k,:}|v_k \in e_i\}, \boldsymbol{E}_{i,:})$$
$$= \boldsymbol{E}'_{i,:}, \forall i \in [|\mathcal{E}|].$$

Representations of virtual node-incidences are

$$\boldsymbol{H}'_{|\mathcal{E}|+j,j,:} = f_{\mathcal{E}\to\mathcal{V}}(\{\boldsymbol{H}_{k,j,:}|v_j \in e_k\}, \boldsymbol{H}_{|\mathcal{E}|+j,j,:})$$
$$= f_{\mathcal{E}\to\mathcal{V}}(\{\boldsymbol{E}'_{i,:}|v_j \in e_i\}, \boldsymbol{X}_{j,:})$$
$$= \boldsymbol{X}'_{j,:}, \forall j \in [|\mathcal{V}|].$$

□

# D DATASET AND BASELINES

The six datasets in the hyperedge-dependent labelling tasks, including Coauth-DBLP, Coauth-AMiner, Email-Enron, Email-Eu, Stack-Biology, and Stack-Physics, are retrieved from Choe et al. (2023). The two Coauth datasets are authors connected by their co-authored publications, where the hyperedge-dependent node labels indicate whether an author is the first author, the last author, or an author in another position within the publication. The two Email datasets are people connected by their exchanged emails, where the hyperedge-dependent node labels specify whether a person is the sender, the receiver, or neither, in the context of the email. In the Stack-Biology and Stack-Physics datasets, the hyperedges denote posts on the stackoverflow.com question-answer platform, with the nodes representing the users who contribute to these posts. The hyperedge-dependent labels indicate whether a user is the one who asked the question, the user whose answer was accepted by the questioner, or another type of contributor. For each dataset, we follow the settings of Choe et al. (2023) to generate 44-dimensional node features $\boldsymbol{X}$, initialized with 2nd-order random walks, and 4-dimensional incidence attributes **B** of positional encodings (i.e. the WithinOrderPE). We summarize their characteristics in the upper part of Table 3. In the lower part of this table, we report the

Table 4: Averaged and standard deviation of Micro-F1 scores (%) across ten runs for node classification on homophilic graphs. The best score for each dataset is **bolded**, the second best is underlined, and the third is *italic*.

| | Pubmed | Cora | Citeseer | DBLP-CA | ModelNet40 | Cora-CA | NTU2012 |
|---|---|---|---|---|---|---|---|
| #nodes | 19,717 | 2,708 | 3,312 | 41,302 | 12,311 | 2,708 | 2,012 |
| #hyperedges | 7,963 | 1,579 | 1,079 | 22,363 | 12,311 | 1,072 | 2,012 |
| #incidences | 34,629 | 4,786 | 3,453 | 99,561 | 61,555 | 4,585 | 10,060 |
| #features | 500 | 1,433 | 3,703 | 1,425 | 100 | 1,433 | 100 |
| #classes | 3 | 7 | 6 | 6 | 40 | 7 | 67 |
| MLP | 87.47±0.51 | 75.17±1.21 | 72.67±1.56 | 84.83±0.22 | 96.14±0.36 | 74.31±1.89 | 85.52±1.49 |
| CEGCN | 86.45±0.43 | 76.17±1.39 | 70.16±1.31 | 88.00±0.26 | 89.92±0.46 | 77.05±1.26 | 81.52±1.43 |
| CEGAT | 86.81±0.42 | 76.41±1.53 | 70.63±1.30 | 88.59±0.29 | 92.52±0.39 | 76.16±1.19 | 82.21±1.23 |
| HNHN | 86.90±0.30 | 76.36±1.92 | 72.64±1.57 | 86.78±0.29 | 97.84±0.25 | 77.19±1.49 | 89.11±1.44 |
| HGNN | 86.44±0.44 | 79.39±1.36 | 72.45±1.16 | 91.03±0.20 | 95.44±0.33 | 82.64±1.65 | 87.72±1.35 |
| HCHA | 86.41±0.36 | 79.14±1.02 | 72.42±1.42 | 90.92±0.22 | 94.48±0.28 | 82.55±0.97 | 87.48±1.87 |
| HyperGCN | 82.84±8.67 | 78.45±1.26 | 71.28±0.82 | 89.38±0.25 | 75.89±5.26 | 79.48±2.08 | 56.36±4.86 |
| UniGCNII | 88.25±0.40 | 78.81±1.05 | 73.05±2.21 | 91.69±0.19 | 98.07±0.23 | 83.60±1.14 | *89.30±1.33* |
| HAN | 86.21±0.48 | 80.18±1.15 | 74.12±1.52 | 90.89±0.23 | 94.04±0.41 | *84.04±1.02* | 83.58±1.46 |
| AllSet | 88.75±0.33 | 78.59±1.47 | 73.08±1.20 | 91.53±0.23 | *98.20±0.20* | 83.63±1.47 | 88.69±1.24 |
| HyperND | 86.68±0.43 | 79.20±1.14 | 72.62±1.49 | 90.35±0.26 | — | 80.62±1.32 | — |
| ED-HNN | **89.56±0.62** | *80.31±1.35* | 73.70±1.38 | **91.93±0.29** | — | 83.97±1.55 | — |
| PhenomNN | 88.25±0.42 | **82.29±1.42** | **75.10±1.59** | 91.91±0.21 | **98.66±0.20** | **85.81±0.90** | **91.03±1.04** |
| **HMP** | *88.45±0.38* | 80.35±1.32 | *73.82±1.21* | 91.87±0.20 | 98.54±0.26 | 84.61±1.35 | 90.70±1.06 |

Table 5: Averaged and standard deviation of Micro-F1 scores (%) across ten runs for node classification on heterophilic graphs. The best score for each dataset is **bolded**, the second best is underlined, and the third is *italic*.

| | Congress | Walmart | House | Senate | 20Newsgroups | Yelp |
|---|---|---|---|---|---|---|
| #nodes | 1,718 | 88,860 | 1,290 | 282 | 16,242 | 50,758 |
| #hyperedges | 83,105 | 6,990 | 341 | 315 | 100 | 679,302 |
| #incidences | 733,994 | 460,630 | 11,843 | 5,408 | 65,451 | 4,523,594 |
| #features | 100 | 100 | 100 | 100 | 100 | 1,862 |
| #classes | 2 | 11 | 2 | 2 | 4 | 9 |
| MLP | — | 45.51±0.24 | 67.93±2.33 | — | *81.42±0.49* | 31.96±0.44 |
| CEGCN | — | 54.44±0.24 | 62.80±2.61 | — | — | — |
| CEGAT | — | 51.14±0.56 | 69.09±3.00 | — | — | — |
| HNHN | 53.35±1.45 | 47.18±0.35 | 67.80±2.59 | 50.93±6.33 | 81.35±0.61 | 31.65±0.44 |
| HGNN | 91.26±1.15 | 62.00±0.24 | 61.39±2.96 | 48.59±4.52 | 80.33±0.42 | *33.04±0.62* |
| HCHA | 90.43±1.20 | 62.45±0.26 | 61.36±2.53 | 48.62±4.41 | 80.33±0.80 | 30.99±0.72 |
| HyperGCN | 55.12±1.96 | 44.74±2.81 | 48.31±2.93 | 42.45±3.67 | 81.05±0.59 | 29.42±1.54 |
| UniGCNII | 94.81±0.81 | 54.45±0.37 | 67.25±2.57 | 49.30±4.25 | 81.12±0.67 | 31.70±0.52 |
| HAN | — | 48.57±1.04 | 71.05±2.26 | — | 79.72±0.62 | 26.05±1.37 |
| HyperGT | 95.23±0.73 | 69.83±0.39 | **74.55±1.99** | 65.49±5.11 | — | — |
| AllSet | 92.16±1.05 | 65.46±0.25 | 69.33±2.20 | 51.83±5.22 | 81.38±0.58 | **36.89±0.51** |
| HyperND | 74.63±3.62 | 38.10±3.86 | 51.70±3.37 | 52.82±3.20 | — | — |
| ED-HNN | *95.19±1.34* | *67.24±0.45* | *73.95±1.97* | *64.79±5.14* | — | — |
| PhenomNN | — | 64.11±0.49 | 71.77±1.68 | — | **81.74±0.52** | 32.26±0.40 |
| **HMP** | **95.77±1.15** | **72.42±0.46** | 74.09±1.95 | **68.31±5.32** | 81.64±0.44 | 36.48±0.44 |

detailed scores, also retrieved from Choe et al. (2023), of HNHN, HGNN, HCHA, HAT, UniGCNII, HNN, HST, and AllSetTransformer, which construct the '8 Baselines' in Table 1.

For node classification, we summarize the datasets and baselines in Table 4 and Table 5. Baselines are from recent literature (Chien et al., 2021; Wang et al., 2023a;b; Liu et al., 2023b), including GCN and GAT on the clique expansion (CEGCN and CEGAT), HNHN (Dong et al., 2020), HCHA (Bai et al., 2021), HyperGCN (Yadati et al., 2019), HyperND (Tudisco et al., 2021), and those we mentioned in Table 2.

# E  ABLATION STUDY

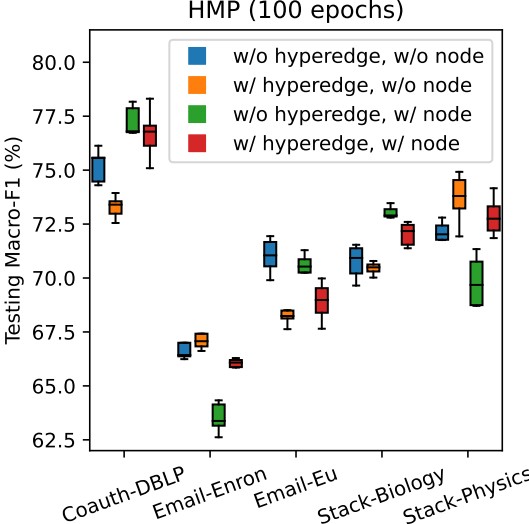

Figure 6: Box plots of Macro-F1 scores across 5 runs for the hyperedge-dependent labelling tasks with/without virtual hyperedge-/node-incidences.

This section illustrates the impact of virtual incidences in HMP using the Macro-F1 scores of HMP with 100 epochs of training in the hyperedge-dependent labelling experiments. Figure 6 depicts HMP's performance when different types of virtual incidences are involved. As it reveals, the use of virtual hyperedge-incidences proves advantageous for tasks involving the Email-Enron and Stack-Physics datasets. Conversely, virtual node-incidences yield better results on the Coauth-DBLP and Stack-Biology datasets. These findings suggest that, even on incidence-level tasks where hyperedge- or node-level representations are not explicitly required, the choice of virtual incidences still has a pronounced effect on performance across different datasets.

# F  HMP HYPERPARAMETERS

## F.1  SENSITIVITY ANALYSIS

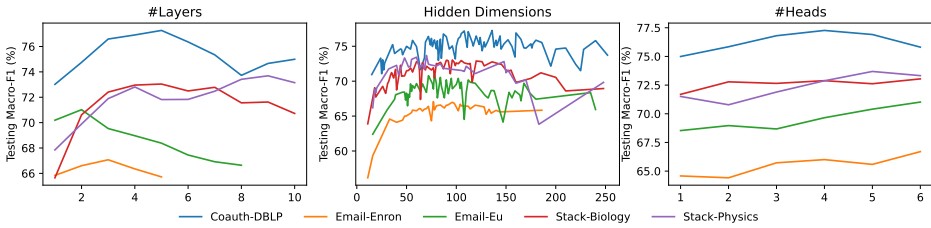

Figure 7: The Macro-F1 score (%) of HMP (100 epochs) with different hyperparameters.

Figure 7 shows the impacts of hyperparameters (i.e., the number of layers, hidden dimensions, and attention heads) on the performance of HMP. As it illustrates, although residual connections are designed in the HMP algorithm, its performance is still sensitive to the number of layers. For different datasets, the optimal performance has to be achieved by tuning the number of layers to an appropriate value. The model performance is less sensitive to the hidden dimensions of the representation, but a sufficient number of hidden dimensions is required to ensure an adequate amount of learnable parameters. Within the hyperparameter range of the experiments, the model generally performs better with a larger number of attention heads, which helps capture complex high-order information in hypergraphs.

## F.2   FINE TUNING

The hyperparameters for HMP in hyperedge-dependent labelling and node classification are optimized using Optuna (Akiba et al., 2019), with the following search space: the number of layers ranging from 1 to 10, the number of attention heads ranging from 1 to 8, and the hidden dimensions for each head ranging from 8 to 64. For hyperedge-dependent labelling, the dropout rate is searched within the range from 0 to 0.2. For node classification, the dropout rate is searched within the range from 0 to 0.8.

## G   EFFICIENCY ANALYSIS

Table 6: The 'script time (in seconds) / peak GPU memory (in MB)' on the Hyperchains datasets.

| Method | #Params | width=2 | width=4 | width=6 | width=8 | width=10 |
|--------|---------|---------|---------|---------|---------|----------|
| AllSet | 49028 | 222 / 542 | 389 / 946 | 535 / 1363 | 693 / 1760 | 856 / 2169 |
| **HMP** | 50885 | 333 / 855 | 482 / 1710 | 667 / 2565 | 826 / 3392 | 1076 / 4289 |
| ED-HNN | 50055 | 302 / 1716 | 584 / 3392 | 863 / 5120 | 1134 / 6776 | 1424 / 8481 |

HMP uses self-attention as its message exchanger and leverages existing advancements in attention mechanisms (Choromanski et al., 2021; Luo et al., 2025) to improve efficiency. To analyze the runtime and memory usage of HMP on hypergraphs with varying hyperedge sizes, we train AllSet (AllDeepSets), ED-HNN, and HMP (with one attention head) for 1000 epochs on the Hyperchains datasets of 5 classes with a fixed length of 10 and varying widths (and thus varying hyperedge sizes). All models have 10 layers and adaptive hidden dimensions, resulting in approximately 50k learnable parameters. The time and memory consumptions with a NVIDIA GeForce RTX 3060 GPU are reported in Table 6. Specifically, except for the highest time consumption on 2-walk hyperchains, HMP's time and space usage fall between AllSet and ED-HNN as the width increases, demonstrating moderate practical efficiency. Overall, the ratio of time and space consumption between HMP and AllSet remains stable within 2 as the hyperedge size (width) increases, validating our complexity analysis that HMP, after efficiency improvements, has the same complexity as AllSet.

