# OpenReview forum: "Hypergraph-Native Message Passing: An Incidence-Centric Perspective"
_ICLR.cc/2026/Conference — Submitted to ICLR 2026_

### Official Review · Reviewer_RCRj · 2025-10-29

**Soundness:** 2
**Presentation:** 3
**Contribution:** 2
**Rating:** 2
**Confidence:** 4

**Summary:**

This paper introduces the Hypergraph-native Message Passing (HMP) framework for learning on hypergraphs. The central motivation is to overcome the "information squashing" problem that affects many existing Hypergraph Neural Networks (HNNs) based on the star expansion. The authors substantiate the framework's efficacy with both theoretical proofs and empirical validation.

**Strengths:**

1.	The manuscript is well-written. The intuition of the "information squashing" problem is explained clearly, which makes the paper's goals easy to understand.

2.	Experimental validation demonstrates that the proposed method can outperform many baseline methods on the selected benchmarks.

**Weaknesses:**

1.	My primary concern is regarding the claimed novelty of the "incidence-centric" idea, which appears to have significant overlap with established previous works. Specifically, the core concept strongly resembles the line expansion (LE) approach [1] and, even more so, the co-representation learning (CoNHD) framework [2]. The paper's dismissal of LE for treating the converted graph as "homogeneous" seems superficial, as LE constructs a well-defined graph of (vertex, hyperedge) pairs with a clear structure. Furthermore, HMP shares nearly identical motivations and high-level solutions with CoNHD, which also models interactions as "multi-input multi-output functions" to avoid information loss. The paper's main attempt to distinguish itself, by critiquing CoNHD's specific SetTransformer implementation as less adaptive, focuses on a low-level implementation choice rather than a fundamental difference in the learning paradigm. This positions HMP less as a novel framework and more as an alternative (e.g., self-attention-based) implementation of the CoNHD paradigm, which diminishes the claimed conceptual contribution.

2.	Following the first point, the paper's central claim of achieving "Adaptive Dimensions" as a key advantage over CoNHD is vague and not well-substantiated. The authors contend that their self-attention mechanism is inherently more adaptive to hyperedge size than CoNHD's SetTransformer. However, this argument seems flawed, as standard multi-head self-attention (used in HMP) also aggregates inputs into a fixed-size output representation, with its 'adaptiveness' coming from learned attention weights. This weighting principle is also central to the SetTransformer. It is therefore unclear how HMP offers a fundamentally more adaptive representation capacity, and this core claimed benefit remains unconvincing.

3.	The empirical analysis feels incomplete and would be strengthened by addressing two key omissions. First, while Theorem 2 commendably identifies the high computational complexity per hyperedge, the proposed solution, referencing efficient transformers, is presented without any empirical validation. The paper would be much more convincing if it included experiments on runtime and scalability to demonstrate the practical viability of this approach. Second, the paper reports that HMP underperforms on key homophilic benchmarks (Cora, Citeseer, Pubmed) but then offers no analysis for this important negative result. A detailed discussion is required to explain why the proposed message-passing mechanism struggles in these standard settings, as this insight is crucial for understanding the model's limitations and applicability.

[1] Yang, C., Wang, R., Yao, S., & Abdelzaher, T. (2022, October). Semi-supervised hypergraph node classification on hypergraph line expansion. In Proceedings of the 31st ACM international conference on information & knowledge management (pp. 2352-2361).

[2] Zheng, Y., & Worring, M. (2024). Co-Representation Neural Hypergraph Diffusion for Edge-Dependent Node Classification. arXiv preprint arXiv:2405.14286.

**Questions:**

1.	Could you clarify the fundamental theoretical difference between HMP's learning paradigm and that of LE and CoNHD, beyond the specific implementation choice?

2.	Given that both multi-head self-attention and SetTransformers aggregate inputs to a fixed-size output, what is the specific mechanism in HMP that enables a fundamentally more "adaptive" representation capacity?

3.	Can you provide the empirical runtime and scalability experiments that validate your claim that efficient transformers are a practical solution to the complexity identified in Theorem 2?

4.	What is your analysis for why the proposed HMP message-passing mechanism underperforms on key homophilic benchmarks like Cora, Citeseer, and Pubmed?

5.	Could you elaborate on how your critique of LE as "homogeneous" holds, given its well-defined bipartite structure of (vertex, hyperedge) pairs?

---

> ### Author Response · Authors · 2025-11-14
>
> Thank you very much for your comments and thorough evaluation of our work.
>
> In A1 and A2 of our common responses, we analyzed HMP's advantages over LE and CoNHD, respectively. Specifically, LE fails to distinguish between edge types among incidences, resulting in the loss of some hypergraph structural information. This is addressed by HMP's second innovation: learning on hypergraph duals, a key component of our proposed incidence-centric learning paradigm. While CoNHD shares the same motivation, its implementation introduces inducing points that create new information bottlenecks, reverting its method to the message passing on star expansion framework and failing to truly realize an incidence-centric solution. We hope this addresses your Q1 and Q5 and resolves your W1.
>
> Regarding adaptiveness, HMP and CoNHD share the same motivation. Both recognize that traditional message passing methods (from the node perspective, as shown in Figure 1b of our manuscript) follow the path "neighbouring nodes -> special hyperedge node -> neighbouring nodes". HMP's analysis identifies the intermediate special hyperedge node as a bottleneck, while CoNHD's analysis notes that the special hyperedge node has a fixed-sized representation, unable to provide adaptive-sized representations for varying numbers of neighbouring nodes (equal to hyperedge size). These analyses are consistent, leading both to propose using multi-input multi-output functions to process each incidence's information and generate personalized representations for each incidence, thereby resolving the bottleneck and achieving adaptiveness. However, CoNHD's implementation effectively replaces the message passing path with "neighbouring nodes -> k inducing points -> neighbouring nodes", where the "k inducing points" act as a special hyperedge node with a representation size expanded k-fold. This essentially undermines CoNHD's claimed adaptiveness. We hope this addresses your Q2 and resolves your W2.
>
> In A4 of our common responses, we added experiments and analyses on HMP's efficiency across different hyperedge sizes, showing that HMP's actual time and space consumption falls between AllSet and ED-HNN, with a stable ratio relative to AllSet. This validates our claim that HMP, after efficiency improvements, has the same complexity as AllSet. We hope this addresses your Q3 and resolves the relevant concerns in your W3.
>
> Regarding the negative results on homophilic benchmarks, we respond in A3 of our common responses. HMP underperforms in node classification tasks on homophilic hypergraphs (Cora, Citeseer, Pubmed) because baselines like ED-HNN and PhenomNN adopt diffusion-based propagation rules, whose inherent homophilic inductive bias aligns well with the structural characteristics of these datasets. Furthermore, additional evidence shows that while baselines with homophilic inductive bias may outperform HMP on small-scale homophilic hypergraphs, this advantage is offset by HMP's strong learning ability as data scale increases. Meanwhile, HMP maintains superior performance on heterophilic hypergraphs, demonstrating its broader applicability. We hope this addresses your Q4 and resolves the relevant concerns in your W3.
>
> Please let us know if any explanation is insufficient. Thank you for your valuable efforts again.

---

### Official Review · Reviewer_a6qN · 2025-10-30

**Soundness:** 3
**Presentation:** 3
**Contribution:** 3
**Rating:** 8
**Confidence:** 4

**Summary:**

This paper proposes an interesting way of thinking for hypergraph learning.  Instead of the usual star expansion, hypergraph native message passing is introduced to pass messages directly between node hyperedge pairs, to avoid information squashing, which is further shown to be generalisation of prior work like AllSet, with gains on extensive tasks.

**Strengths:**

The core idea is interesting. The incidence-centric perspective is more natural fit for hypergraphs. The experiment results are promising.

**Weaknesses:**

The theory says complexity scales with the size of the largest hyperedge, what will happen in practice with a dataset that has massive hyperedges? A short discussion or experiment on this would be more convincing.

**Questions:**

Could you provide practical guidance on when to use virtual node incidences or hyperedge incidences?

Will the model performance degreade as the hyperedge become extremely large and if there is a point that self attention within a hyperedge become the bottleneck?

---

> ### Author Response · Authors · 2025-11-14
>
> Thank you very much for your comments and kind words.
>
> Our ablation studies (Figure 6) indicate that there are no universal guidelines for choosing virtual incidences, and tuning the four combinations (with/without virtual hyperedge-/node-incidences) remains necessary. However, such hyperparameter tuning is standard in deep learning, and we hope this does not overshadow the core contributions of our incidence-centric framework.
>
> For extremely large hyperedges, HMP adaptively uses Performer [1], a linear-complexity attention implementation, to ensure efficiency, while retaining the original quadratic-complexity self-attention for other normal hyperedges to maintain accuracy. Thus, extreme hyperedges do not become efficiency bottlenecks.
>
> Concrete evidence comes from our experiments on Email-Eu, which contains 986 nodes and 541,842 incidences. Its dual hypergraph includes extremely large hyperedges due to highly connected nodes. HMP ran efficiently on this dataset (no out-of-memory errors, with training time comparable to baselines) while achieving state-of-the-art performance, confirming that extreme hyperedges do not pose efficiency issues.
>
> Please let us know if any explanation is insufficient. We sincerely thank you again.
>
> > [1] Krzysztof, C., Valerii, L., David, D., Xingyou, S., Andreea, G., Tamas, S., ... & Adrian, W. (2021). Rethinking attention with performers. Proceedings of ICLR.

---

### Official Review · Reviewer_N534 · 2025-10-31

**Soundness:** 3
**Presentation:** 3
**Contribution:** 3
**Rating:** 6
**Confidence:** 2

**Summary:**

The paper introduces Hypergraph-native Message Passing (HMP), a novel framework for learning on hypergraphs. The authors argue that existing Hypergraph Neural Networks (HNNs), which often adapt graph neural network (GNN) paradigms, suffer from information squashing. HMP proposes an incidence-centric perspective, instead of the traditional 'node-hyperedge-node' path.

**Strengths:**

The paper clearly identifies a key limitation in existing HNNs, i.e., the "information squashing" bottleneck. Doing the same kind of incidence exchange on the dual hypergraph is elegant and makes the framework naturally handle hyperedge-centric signals as well. The synthetic hyperchains are set up specifically to test whether a method can preserve higher-order paths without over-squashing. HMP is indeed the most robust

**Weaknesses:**

The primary concern is computational cost. While they mention using linear-complexity attention and parallelization, this feels like a partial solution. For hypergraphs with very large hyperedges, this quadratic cost within every hyperedge, every layer, could be prohibitive compared to the simple aggregation in methods like AllSet.

**Questions:**

Could the authors elaborate on the practical runtime/memory trade-offs?

---

> ### Author Response · Authors · 2025-11-14
>
> Thank you very much for your thoughtful comments.
>
> In A4 of our common responses, we added experiments and analyses on HMP's efficiency across different hyperedge sizes, showing that HMP's actual time and space consumption falls between AllSet and ED-HNN, with a stable ratio relative to AllSet. This validates our claim that HMP, after efficiency improvements, has the same complexity as AllSet.
>
> Theoretically, for very large hyperedges, HMP adaptively uses Performer [1], a linear-complexity attention implementation, to ensure efficiency, while retaining the original quadratic-complexity self-attention for other normal hyperedges to maintain accuracy. Thus, very large hyperedges do not become efficiency bottlenecks.
>
> Concrete evidence comes from our experiments on Email-Eu, which contains 986 nodes and 541,842 incidences. Its dual hypergraph includes very large hyperedges due to highly connected nodes. HMP ran efficiently on this dataset (no out-of-memory errors, with training time comparable to baselines) while achieving state-of-the-art performance, confirming that very large hyperedges do not pose efficiency issues.
>
> With sincere thanks, we hope this addresses your questions and resolves your concerns. Please let us know if any explanation is insufficient.
>
> > [1] Krzysztof, C., Valerii, L., David, D., Xingyou, S., Andreea, G., Tamas, S., ... & Adrian, W. (2021). Rethinking attention with performers. Proceedings of ICLR.

---

### Official Review · Reviewer_2FbW · 2025-11-01

**Soundness:** 3
**Presentation:** 3
**Contribution:** 2
**Rating:** 4
**Confidence:** 3

**Summary:**

This paper proposes the Hypergraph-native Message Passing (HMP) framework, a novel approach for representation learning on hypergraphs. The key motivation is to address the "information squashing" problem prevalent in existing Hypergraph Neural Networks (HNNs), which often rely on a star-expansion paradigm where messages from multiple nodes are aggregated into a single, fixed-size hyperedge representation. This bottleneck is particularly detrimental in large or heterophilic hyperedges. The authors demonstrate both theoretical and empirical results to support the effectiveness of the proposed HMP framework.

**Strengths:**

- The manuscript is clearly written and logically organized.
- The proposed method demonstrates strong empirical performance, outperforming many baseline methods on the selected benchmarks.
- The paper provides a useful theoretical contribution, offering theorems that prove HMP can be seen as a generalization of AllSet and, by extension, many other HNNs.

**Weaknesses:**

- **Incremental Novelty:** The paper's claim to propose a "pioneering learning paradigm" is unsubstantiated. The core "incidence-centric" idea is predated by frameworks like Line Expansion (LE) [1], which is not adequately discussed. HMP's novelty over LE appears to be the application of self-attention, not the paradigm itself. Similarly, the paper claims novelty over CoNHD [2] by handling adaptive hyperedge representations, but this overlooks the fact that adaptive representation sizes are already a key contribution of CoNHD. Consequently, the novelty appears incremental.
- **Insufficient Motivation:** The paper is motivated as a solution to the "information squashing" problem of HNNs, but this issue is primarily demonstrated in the context of star-expansion. The paper fails to establish that this problem even exists in more relevant, related works like LE and CoNHD. If these prior methods already resolve information squashing, the paper's core motivation is not strong enough.
- **Lack of Theoretical Analysis:** Given that related works (LE, CoNHD) also appear to mitigate the information squashing problem, the paper must provide a more rigorous theoretical analysis to differentiate its contribution. The authors should formally prove what advantages HMP offers in terms of expressive power or other theoretical properties when compared directly against these existing methods.
- **Insufficient Empirical Analysis:** The paper's empirical analysis is incomplete. It reports that HMP underperforms on key homophilic benchmarks (Cora, Citeseer, Pubmed) but offers no analysis for this important negative result. An analysis is required to explain why HMP's message-passing mechanism fails in these settings. For example, the paper should investigate the trade-off between HMP's flexible attention and the strong, beneficial smoothing bias of the baseline methods that outperform it.

[1] Yang et al., "Semi-supervised hypergraph node classification on hypergraph line expansion,” Proceedings of the 31st ACM international conference on information and knowledge management, 2022.

[2] Zheng and Worring, ”Co-Representation Neural Hypergraph Diffusion for Edge-Dependent Node Classification,” arXiv, 2024.

**Questions:**

- Could you please clarify the theoretical difference between the proposed 'incidence-centric' paradigm and the Line Expansion (LE) framework [1]?
- Given the similarity to LE, how would you redefine HMP's core novelty beyond the specific application of self-attention?
- Since CoNHD [2] also features an adaptive representation mechanism, could you provide a direct comparison to demonstrate what unique advantages HMP's mechanism offers?
- Can you provide theoretical or empirical evidence that the information squashing problem persists in more advanced frameworks like LE and CoNHD?
- Could you provide a formal analysis of expressive power that theoretically establishes the advantages of HMP's self-attention mechanism over the schemes used in LE and CoNHD?
- Could you provide a detailed analysis for the negative result that HMP underperforms on key homophilic benchmarks like Cora and Citeseer?

[1] Yang et al., "Semi-supervised hypergraph node classification on hypergraph line expansion,” Proceedings of the 31st ACM international conference on information and knowledge management, 2022.

[2] Zheng and Worring, ”Co-Representation Neural Hypergraph Diffusion for Edge-Dependent Node Classification,” arXiv, 2024.

---

> ### Author Response · Authors · 2025-11-14
>
> Thank you very much for your valuable efforts on reviewing our manuscript.
>
> Contrary to your understanding, the difference between HMP and LE lies not in the application of self-attention but in the paradigm itself. For a detailed analysis, please refer to A1 in our common responses. In summary, LE fails to distinguish between edge types among incidences, resulting in the loss of some hypergraph structural information. In contrast, HMP retains and utilizes this structural information through its second innovation, learning on hypergraph duals, thereby achieving higher expressiveness. We hope this addresses your Q1, Q2, and the parts of Q4 and Q5 related to LE, and partially resolves your W1, W2, and W3.
>
> Regarding CoNHD, we provide a more detailed analysis in A2 of our common responses. Briefly, during within-edge interactions (and similarly for within-node interactions), CoNHD's implementation introduces multiple inducing points into the hypergraph to replace the "special hyperedge-node" in message passing on star expansion (MP on SE). While this widens the hyperedge bottleneck and alleviates the information squashing issue, the same effect can be achieved by expanding the representation of the special hyperedge-node in MP on SE. Thus, its expressive power does not exceed that of AllSet, and consequently, HMP. We hope this addresses your Q3, the parts of Q4 and Q5 related to CoNHD, and partially resolves your W1, W2, and W3.
>
> Regarding the negative results on homophilic benchmarks, we respond in A3 of our common responses. HMP underperforms in node classification tasks on homophilic hypergraphs (Cora, Citeseer, Pubmed) because baselines like ED-HNN and PhenomNN adopt diffusion-based propagation rules, whose inherent homophilic inductive bias aligns well with the structural characteristics of these datasets. Furthermore, additional evidence shows that while baselines with homophilic inductive bias may outperform HMP on small-scale homophilic hypergraphs, this advantage is offset by HMP's strong learning ability as data scale increases. Meanwhile, HMP maintains superior performance on heterophilic hypergraphs, demonstrating its broader applicability. We hope this addresses your Q6 and resolves your W4.
>
> Please let us know if any explanation is insufficient. Thank you again.

---

> ### Comment · Reviewer_2FbW · 2025-11-27
>
> Many thanks to the authors for the rebuttal. After carefully reviewing the response and the comments from the other reviewers, I have decided to maintain my current score. The key reason remains that the contribution is incremental compared with line expansion (LE) and CoNHD. While CoNHD can be regarded as concurrent work, the central claimed novelty, namely the incidence-centric perspective, still appears too close to LE.
>
> In my view, the statement in the rebuttal that “LE fails to distinguish between incidences that share nodes and those that share hyperedges, leading to irreversible loss of hypergraph structural information during message passing” is debatable. LE is a bijective hypergraph transformation, so it preserves all structural information at the level of topology. Any apparent information “loss” is therefore not inherent to LE itself but stems from model design: a naïve GCN on LE that treats all edges uniformly may ignore contextual structure, whereas a more sophisticated message-passing scheme can exploit the implicit structure (including the mechanism proposed in this work).
>
> Overall, I see the proposed method more as a follow-up to LE: the attention mechanism differentiates it from a vanilla LE-GCN, but the incidence-centric message passing illustrated in Figure 2 (c) is, in essence, equivalent to performing message passing on the LE-transformed hypergraph.

---

> > ### Author Response · Authors · 2025-11-27
> >
> > Thank you for giving us this opportunity to provide further clarification.
> >  We would be honored to first delve into the Line Expansion (LE) method with you.
> >
> > Assuming a hypergraph has $n$ vertices, $m$ hyperedges, and $b$ incidences, according to [the 7th paragraph in the Introduction section of the LE paper](https://arxiv.org/pdf/2005.04843) and ["Step 2" in the README.md file of the official LE repository](https://github.com/ycq091044/LEGCN/tree/master?tab=readme-ov-file#step-2-run-your-graph-algorithm), the LE method consists of the following four steps:
> >
> > 1. Transform the hypergraph into an LE graph, deriving a $b \times b$ adjacency matrix $A$ and four projection matrices: $P_v$ ($b \times n$), $P_v^T$ ($n \times b$), $P_e$ ($b \times m$), and $P_e^T$ ($m \times b$);
> > 2. Project vertex features to incidences using $P_v$, obtaining incidence features $X$;
> > 3. Apply a GNN on the graph defined by $A$ and $X$ to obtain incidence representations $H$;
> > 4. Project $H$ back to vertices using $P_v^T$.
> >
> > Thus, your statement that "LE is a bijective hypergraph transformation" is indeed correct because Theorem 1 in the LE paper proves that the original hypergraph can be recovered using the complete set of information including $A$, $P_v$, $P_v^T$, $P_e$, and $P_e^T$.
> > **However**, a critical observation arises when applying the GNN.
> > As described in Step 3 above, the GNN only receives inputs $A$ and $X$ during the message-passing process.
> > The adjacency matrix $A$ is homogeneous and lacks the structural knowledge to distinguish between incidences that share vertices and those that share hyperedges.
> > This specific structural information is exclusively stored in the projection matrices $P_v$, $P_v^T$, $P_e$, and $P_e^T$, which are not fed into the GNN.
> >
> > Therefore, our previous statement regarding the limitation of LE remains valid, unless you:
> >
> > 1. design a specialized GNN to incorporate the projection matrices $P_v$, $P_v^T$, $P_e$, and $P_e^T$. However, this falls beyond the scope of the original LE method, as the LE paper explicitly claims that LE "enables **the existing graph learning algorithms** to work";
> > 2. consider Step 2 to 4 as an integrated "LE layer". However, in this way, the repeated projection of incidence information to vertices and back creates inevitable vertex-level bottlenecks, and this is precisely the limitation that our method overcomes.
> >
> > In conclusion, our proposed method can truly support the incidence-centric perspective through its effective handling of hypergraph duality, without losing structural knowledge during message passing or introducing information bottlenecks.
> >
> > Hoping this can change your decision. Thank you.

---

### Author Response · Authors · 2025-11-14
**A1. Common Response: Detailed Differences Between HMP and Line Expansion (LE) [1]**

We note that "LE treats the converted graph as homogeneous" (as stated in line 294 of our Related Works) is directly supported by LE's original description, specifically the third paragraph in the Introduction section of [1]:

> We propose line expansion (LE) for hypergraphs, which is a powerful bijective mapping from a hypergraph structure to a **homogeneous** graph structure.

The reason for this homogeneity is explained in the fourth paragraph of [1]:

> “edge” between two "node"s are constructed if two "node"s share the same vertex **or** hyperedge.

To illustrate, consider Figure 2c in our manuscript, which uses incidences as nodes. The red dashed lines (between incidences sharing hyperedges) and black solid lines (sharing nodes) are two distinct types of edges. Recent methods such as AllSet, CoNHD [2], and our proposed HMP all distinguish between these two edge types, for example, by using different rules to propagate information from incidence c2 to incidences c1 and a2, respectively. However, LE fails to distinguish between incidences that share nodes and those that share hyperedges, leading to irreversible loss of hypergraph structural information during message passing. (This structural information is stored in an additional vertex projection matrix but does not participate in message passing.) This is also reflected in our experimental results on Hyperchains, where the LEGCN model performs the worst.

> * [1] Yang, C., Wang, R., Yao, S., & Abdelzaher, T. (2022, October). Semi-supervised hypergraph node classification on hypergraph line expansion. In Proceedings of the 31st ACM international conference on information & knowledge management (pp. 2352-2361).
> * [2] Zheng, Y., & Worring, M. (2024). Co-Representation Neural Hypergraph Diffusion for Edge-Dependent Node Classification. arXiv preprint arXiv:2405.14286.

---

> ### Comment · Reviewer_RCRj · 2025-11-14
>
> Thank you for the rebuttal. I acknowledge that the proposed method has unique components compared to LE [1] and CoNHD [2].
>
> However, my primary concern remains regarding the claimed novelty of the core mechanism. The "Incidence-Centric Message Passing" pipeline in Section 3.1, presented as the key component for the information squashing problem, still shares significant and fundamental overlap with these prior works. Specifically, the pipeline visualised in Figure 2 (c) closely mirrors the message passing scheme conducted on the line expansion, as illustrated in Figure 2 of [1]. Given this strong similarity, the paper's current positioning significantly overstates its contribution. Claims such as being a "pioneering learning paradigm" for the information squashing problem within HNNs like AllSet are not well-supported when viewed against [1] and [2] (even if [2] is treated as concurrent work).
>
> The paper, therefore, requires substantial revision to frame its novelty accurately. A more promising and defensible contribution would be to focus on how the specific attention mechanism mentioned in Section 3.2 (adapted from [3]) enhances or addresses the limitations of a vanilla line expansion (LE) method. Reframing the paper around this aspect would clarify the distinct contribution.
>
> [1] Yang, C., Wang, R., Yao, S., & Abdelzaher, T. (2022, October). Semi-supervised hypergraph node classification on hypergraph line expansion. In Proceedings of the 31st ACM international conference on information & knowledge management (pp. 2352-2361).
>
> [2] Zheng, Y., & Worring, M. (2024). Co-Representation Neural Hypergraph Diffusion for Edge-Dependent Node Classification. arXiv preprint arXiv:2405.14286.
>
> [3] Vaswani, A., Shazeer, N., Parmar, N., Uszkoreit, J., Jones, L., Gomez, A. N., ... & Polosukhin, I. (2017). Attention is all you need. Advances in neural information processing systems, 30.

---

> > ### Author Response · Authors · 2025-11-14
> >
> > Thank you for your feedback.
> >
> > We understand your concern about whether HMP’s core contribution is sufficiently significant given the prior existence of "Incidence-Centric Message Passing" methods like LE.
> >
> > First, we wish to clarify that the title of our Section 3.2 is “Incidence-Centric Learning on Hypergraph **Duals**,” and **duality** is a key factor that distinguishes HMP from and gives it an advantage over LE (detailed reasons are already provided in A1). Throughout our manuscript, whenever we refer to HMP as incidence-centric, we consistently emphasize the **duality**. This is reflected in the Abstract, Introduction, and even Figure 2c you mentioned. Thus, "Incidence-Centric" is not an isolated innovation nor the entirety of our contributions. Any similarity in this aspect to existing work does not negate the innovativeness of our paper.
> >
> > Second, we compare the content of the three subsections in our Methodology to illustrate the differences between our contributions and LE:
> >
> > 1. Section 3.1: We propose HMP from the perspective of addressing the information bottleneck and explain the mechanism behind its success, which is entirely different from LE’s motivation.
> > 2. Section 3.2: We refine HMP through the lens of duality, representing an essential distinction from LE’s homogeneous approach.
> > 3. Section 3.3: While handling incidence-level tasks, we extend HMP to node-level tasks, resulting in a broader scope of application compared to LE, which is limited to node-level tasks.
> >
> > In summary, HMP differs from LE comprehensively in terms of its underlying motivation, learning paradigm, and application scope. The current manuscript frame already clearly reflects the distinctions from LE.
> >
> > By the way, we sincerely thank you for your encouraging suggestion to reframe the manuscript to focus on the attention mechanism. However, to be honest, we respectfully think this idea is not that good, as LE can integrate any GNN after its expansion, and thus it can integrate Graph Attention Networks (GAT)[1] to have its attention mechanism.
> >
> > Please let us know if you require further clarification.
> >
> > > * [1] Velickovic, P., Cucurull, G., Casanova, A., Romero, A., Lio, P., & Bengio, Y. (2017). Graph attention networks. stat, 1050(20), 10-48550.

---

> > ### Author Response · Authors · 2025-11-27
> >
> > Dear Reviewer RCRj,
> > We are having a debate with Reviewer 2FbW on this specific concern about LE.
> > It is our pleasure if you are willing to join us.
> > See https://openreview.net/forum?id=eRu0UBXEh2&noteId=eNSFzND9MQ

---

### Author Response · Authors · 2025-11-14
**A2. Common Response: Detailed Differences Between HMP and CoNHD [2]**

HMP and CoNHD are independent concurrent works. HMP breaks through the fixed-dimensional limitation of CoNHD through dynamic self-attention and constructs a complete research system covering "mechanism-framework-extension".

**First**, HMP and CoNHD are concurrent works. CoNHD was accepted by CIKM 2025 and released its source code in August this year, remaining in arXiv preprint status before that. HMP was submitted to KDD in February this year with simultaneous open-sourcing of code (as evidenced by the anonymous repository in our abstract, where our complete code was made open-source as early as February 13, 2025), at which point we were unaware of the preprint CoNHD. Later, following reviewer suggestions, we added comparisons with CoNHD and extensive theoretical analyses before submitting to ICLR 2026. Thus, the research timelines and code release dates of both works confirm that HMP's core ideas were independently developed. The overlap with CoNHD is a natural coincidence in academic exploration and does not affect its innovativeness.

**Second**, the self-attention mechanism in HMP is not a trivially replaceable module; its implementation is crucial to the viability of the incidence-centric idea, whereas CoNHD's implementation (e.g., ISAB) is insufficient to support this idea. CoNHD's implementation details can be found in its [Appendix C](https://arxiv.org/pdf/2405.14286v2):

> ISAB utilizes a fixed number of inducing points $\mathbf{I} \in \mathcal{R}^{k \times d}$ to reduce the quadratic complexity in self-attention to linear complexity.

ISAB is formally expressed as:

$$\begin{aligned}
\text{ISAB}(X) &= \text{MAB}(X, \text{MAB}(\mathbf{W}^I, X)) \\\\
\text{MAB}(Q, K) &= \text{LN}(M + \text{REF}(M)) \\\\
M &= \text{LN}(Q, \text{MultiHead}(Q, K, K))
\end{aligned}$$

If we regard the inducing points $\mathbf{I}$ as $k$ added nodes in the hypergraph, $\text{MAB}(\mathbf{W}^I, X)$ propagates information from the considered nodes (with $X$) to the $k$ inducing nodes, and the second MAB in ISAB propagates information back from the $k$ inducing nodes to the considered nodes, resulting in a message passing paradigm as shown in the following figure (when $k=4$ as illustrated):

![](https://github.com/user-attachments/assets/7df9a0cc-0424-4dae-b24e-c0670f3296b8)

Here, the first MAB propagates messages from $v_1, v_2, v_3, v_4$ to the $k=4$ inducing nodes $I_1, I_2, I_3, I_4$, and the second MAB propagates messages from $I$ back to $v_1, v_2, v_3, v_4$. As noted in our Related Works (line 299), while ISAB alleviates the information bottleneck issue, its use of a fixed number of inducing points (with $k$ being a global hyperparameter, defaulting to only 4 according to its [code](https://github.com/zhengyijia/CoNHD/blob/main/model/CoNHD_ADMM.py#L86)) actually undermines its claimed adaptiveness, as the representation size cannot adjust with hyperedge size. Moreover, message passing on star expansion (MP on SE) methods only require increasing the intermediate representation size (e.g., expanding the representation size of node $a$ in Figure 1b to match the $k \times d$ dimension of inducing points) to achieve results comparable to CoNHD with ISAB. Thus, CoNHD with ISAB is not more expressive than AllSet or HMP. In summary, although CoNHD, like HMP, identifies the information squashing problem in MP on SE, its implementation still reverts to the MP on SE framework. In contrast, HMP truly realizes an incidence-centric solution, achieving significantly better performance than CoNHD in incidence-level tasks (Table 1).

**Third**, unlike CoNHD, which focuses on simply solving hyperedge-dependent labelling tasks, HMP conducts comprehensive research around the incidence-centric idea, including:

1. Providing theoretical and empirical support for the rationality of the incidence-centric paradigm through s-walk analysis and the introduction of hyperchain datasets.
2. Innovatively unifying message exchanging modelling on two types of incidence interactions by introducing hypergraph duality ("elegant and natural" — Reviewer N534).
3. Proposing virtual incidences to extend HMP to more downstream tasks (e.g., hyperedge-/node-level tasks) without modifying the HMP model, and demonstrating the broad applicability of the incidence-centric paradigm through node classification experiments.

In conclusion, compared to CoNHD's partial improvements, HMP's research forms a complete logical chain from "why it works" (mechanism) to "how to model it" (framework) and "how to apply it" (extension). It not only addresses key scientific questions in the incidence-centric paradigm but also provides reusable methodologies, representing a systematic contribution to the field.

> * [2] Zheng, Y., & Worring, M. (2024). Co-Representation Neural Hypergraph Diffusion for Edge-Dependent Node Classification. arXiv preprint arXiv:2405.14286.

---

### Author Response · Authors · 2025-11-14
**A3. Common Response: Further Analysis of HMP's Performance on Homophilic Hypergraphs**

As noted in line 473 of our manuscript, HMP underperforms in node classification tasks on homophilic hypergraphs (Cora, Citeseer, Pubmed) because baselines such as ED-HNN and PhenomNN adopt diffusion-based propagation rules, whose inherent homophilic inductive bias aligns well with the structural characteristics of these datasets. In contrast, HMP is designed to adapt to broader scenarios (including heterophily) without relying on such task-specific biases, explaining its competitive advantages in heterophilic node-level tasks.

To elaborate, we verify that HMP can learn homophilic inductive bias from data as the hypergraph scale increases. The table below compares HMP with state-of-the-art methods on datasets with CE Homophily > 0.8:

|              | Citeseer | Cora-CA | Cora  | Pubmed | ModelNet40 | DBLP-CA |
|--------------|----------|---------|-------|--------|------------|---------|
| CE Homophily | 0.893    | 0.803   | 0.897 | 0.952  | 0.853      | 0.869   |
| #incidences  | 3k       | 5k      | 5k    | 35k    | 62k        | 100k    |
| SotA         | 75.10    | 85.81   | 82.29 | 89.56  | 98.66      | 91.93   |
| HMP          | 73.82    | 84.61   | 80.35 | 88.45  | 98.54      | 91.87   |
| $\| \Delta \|$   | 1.28     | 1.2     | 1.94  | 1.11   | 0.12       | 0.06    |

Notably, the performance gap ($|\Delta|$) shrinks significantly with increasing incidence count, decreasing from 1.x to 0.06. This aligns with the empirical law of GNNs on conventional graphs [3]: as dataset scale grows, models with flexible learning capabilities (e.g., GAT) narrow the gap with bias-aligned baselines (e.g., GCN/GraphSAGE) on homophilic data:

|               | Cora      | CiteSeer  | PubMed    | CS        | Photo     | WikiCS    | Computer  | Physics   |
|---------------|-----------|-----------|-----------|-----------|-----------|-----------|-----------|-----------|
| #edges        | 5k        | 5k        | 44k       | 82k       | 119k      | 216k      | 246k      | 248k      |
| GCN/GraphSAGE | **85.10** | **73.14** | **81.12** | **96.38** | **96.78** | 80.69     | 93.99     | **97.46** |
| GAT           | 84.46     | 72.22     | 80.28     | 96.21     | 96.60     | **81.07** | **94.09** | 97.25     |

In summary, baselines with homophilic inductive bias may outperform HMP on small-scale homophilic hypergraphs, but this advantage diminishes as data scale increases due to HMP's strong learning ability. Meanwhile, HMP maintains superior performance on heterophilic hypergraphs, demonstrating its broader applicability.

> * [3] Luo, Y., Shi, L., & Wu, X. M. (2024). Classic gnns are strong baselines: Reassessing gnns for node classification. Advances in Neural Information Processing Systems, 37, 97650-97669.

---

### Author Response · Authors · 2025-11-14
**A4. Common Response: Detailed Efficiency Analysis of HMP**

HMP uses self-attention as its message exchanger and leverages existing advancements in attention mechanisms (e.g., Performer [4]) to improve efficiency. Thus, runtime/memory trade-offs are not the core innovation or primary challenge of this work, but their feasibility has been verified. For example, HMP can stably operate on the large-scale Coauth-AMiner dataset (containing 5 million incidences) while achieving state-of-the-art performance.

To address reviewers' concerns, we analyzed the runtime and memory usage of HMP on hypergraphs with varying hyperedge sizes. We trained AllSet (AllDeepSets), ED-HNN, and HMP (with one attention head) for 1000 epochs on Hyperchains datasets (#classes=5) with a fixed length of 10 and varying widths (and thus varying hyperedge sizes). All models have 10 layers and adaptive hidden dimensions, resulting in approximately 50k learnable parameters. The following table reports the "script time (in seconds) & peak GPU memory (in MB)" using a NVIDIA GeForce RTX 3060 GPU:

|         | #params | width=2    | width=4    | width=6    | width=8     | width=10    |
|---------|---------|------------|------------|------------|-------------|-------------|
| AllSet  | 49028   | 222 & 542  | 389 & 946  | 535 & 1363 | 693 & 1760  | 856 & 2169  |
| **HMP** | 50885   | 333 & 855  | 482 & 1710 | 667 & 2565 | 826 & 3392  | 1076 & 4289 |
| ED-HNN  | 50055   | 302 & 1716 | 584 & 3392 | 863 & 5120 | 1134 & 6776 | 1424 & 8481 |

Specifically, except for the highest time consumption on 2-walk hyperchains, HMP's time and space usage fall between AllSet and ED-HNN as the width increases, demonstrating moderate practical efficiency. Overall, the ratio of time and space consumption between HMP and AllSet remains stable within 2 as the hyperedge size (width) increases, validating our complexity analysis that HMP, after efficiency improvements, has the same complexity as AllSet.

> * [4] Krzysztof, C., Valerii, L., David, D., Xingyou, S., Andreea, G., Tamas, S., ... & Adrian, W. (2021). Rethinking attention with performers. Proceedings of ICLR.

---

### Author Response · Authors · 2025-11-30
**Rebuttal Summary**

Dear AC,

We are writing to respond to the OpenReview (OR) leakage and provide a summary of the rebuttal regarding our proposed method, HMP.

We have received feedback from four reviewers, who assigned rating scores of 2, 4, 6, and 8 respectively.

The reviewer who gave an 8 praised HMP as "a more natural fit for hypergraphs" and further inquired about two key aspects: the parameter tuning experience for virtual incidences in practical applications and the robustness of our method when dealing with extreme hyperedge sizes.
We have addressed both questions in [our response to this reviewer](https://openreview.net/forum?id=eRu0UBXEh2&noteId=95k9urcEzQ).

The reviewer who gave a 6 commended HMP for being "elegant and natural" while expressing concern about whether the complexity issue outlined in our Theorem 2 can be effectively resolved by the linear attention mechanism.
In [our response (A4)](https://openreview.net/forum?id=eRu0UBXEh2&noteId=KemAZ7JofN), we provide our experimental findings which demonstrate that: "With linear attention, HMP's time and space usage fall between those of AllSet[1] and ED-HNN[2], showcasing moderate practical efficiency."
Furthermore, the data trends validate our complexity analysis, confirming that HMP achieves the **same complexity as AllSet**.
We think these results and analysis **resolve the concern** and have updated the revised manuscript by incorporating them into the appendices.

Beyond the aforementioned complexity concern, the core questions raised by the reviewers who gave 4 and 2 are **nearly identical**.
While acknowledging the clear writing style of the paper and its strong experimental performance, both reviewers sought clarification on two key points: **(1)** the reasons behind HMP's underperformance compared to the state-of-the-art (SOTA) methods on the Cora, Citeseer, and PubMed datasets; and **(2)** the distinctions and advantages of HMP relative to LE[3] and CoNHD[4].
Although preliminary explanations and discussions on these two points were **already included in our initial manuscript (Lines 473-475 and 294-301)**, we have supplemented the following detailed responses:

1. [A3](https://openreview.net/forum?id=eRu0UBXEh2&noteId=nt9XUFW3HF) provides an in-depth explanation for HMP's underperformance on certain datasets. Our key conclusion is: "Baselines with a homophilic inductive bias may outperform HMP on small-scale homophilic hypergraphs; however, this advantage diminishes as the data scale expands, thanks to HMP's robust learning capacity. Notably, HMP consistently maintains superior performance on heterophilic hypergraphs, highlighting its broader applicability." As reflected in their subsequent comments, **both reviewers expressed no further doubts** regarding this explanation.
2. [A2](https://openreview.net/forum?id=eRu0UBXEh2&noteId=uO3MCW0N92) clarifies the relationship between HMP and CoNHD as concurrent work. Additionally, it emphasizes that HMP offers more in-depth research contributions (truly realizing an incidence-centric solution that is adaptive and free of bottlenecks) and more comprehensive coverage (encompassing a systematic study of mechanisms, frameworks, and extensions). **Both reviewers have affirmed their acceptance** of this clarification in their follow-up comments.
3. [A1](https://openreview.net/forum?id=eRu0UBXEh2&noteId=Rh9E07w4a8) and [the subsequent response](https://openreview.net/forum?id=eRu0UBXEh2&noteId=eNSFzND9MQ) elaborate on the novelty and advancements of HMP compared to LE, specifically targeting **a misunderstanding shared by both reviewers**. LE's homogeneity constraint leads to the loss of certain structural information, whereas HMP preserves and leverages richer hypergraph structural information during message passing by handling hypergraph duality. The experimental results in related papers further illustrate that this inherent limitation of LE results in a significant performance gap between LE (2022) and more recent methods such as ED-HNN (2023), CoNHD (2025), and HMP (ours). Regrettably, due to the occurrence of the OR leakage, we have not yet received the reviewers' feedback on these responses. Nevertheless, we sincerely hope that our detailed rebuttal will facilitate their deeper understanding of LE, enabling them to conduct a more accurate assessment of HMP's novelty.

We would like to express our sincere gratitude to all reviewers for their valuable feedback on our work and to you (the AC) for your additional efforts in addressing this OR leakage incident.

Yours sincerely,

> * [1] Chien et al., "You are AllSet: A Multiset Function Framework for Hypergraph Neural Networks," ICLR 2022
> * [2] Wang et al., "Equivariant Hypergraph Diffusion Neural Operators," ICLR 2023
> * [3] Yang et al., "Semi-supervised hypergraph node classification on hypergraph line expansion," CIKM 2022
> * [4] Zheng and Worring, "Modeling Edge-Specific Node Features through Co-Representation Neural Hypergraph Diffusion," CIKM 2025

---

### Meta-Review · Area_Chair_gpk4 · 2026-01-05

**Summary:**

This paper was reviewed by 4 knowledgeable reviewers, who raised main concerns about:
1. The novelty of the proposed approach, which appeared incremental w.r.t. LE and CoNHD  (2FbW, RCRj), and some claims not sufficiently substantiated (RCRj).
2. The unclear motivation for the proposed approach, and the benefits it offers over prior work in terms of expressive power or other theoretical properties (2FbW).
3. The experimental validation, which appeared unconvincing with results underperforming prior art on homophilic benchmarks. (2FbW , RCRj)
4. The computational cost of the proposed approach (N534, a6qN, RCRj).

**Reviewer Concerns:**

In their rebuttal, the authors argue for the novelty of the proposed approach, highlighting differences with prior work and in particular LE and CoNHD. The rebuttal considers CoNHD as concurrent work, given the publication date of the paper (Aug 2025) despite the paper being on arXiv since May 2024. Although the reviewers acknowledge that the proposed method has unique components w.r.t. these 2 works, they still view the novelty as incremental given the tight connection to LE and CoNHD, and suggest an in-depth analysis on the expressive power or other theoretical properties to emphasize the utility of the contribution and support the claims made. The AC agrees that, although the similarities between the proposed approach and prior work can be explained as a natural coincidence in academic exploration, they also beg for a clear positioning of the work and convincing evidence on the benefits of the proposed approach.

The rebuttal addresses the performance in node classification tasks on homophilic hypergraphs by highlighting the usage of diffusion-based propagation rules in baselines; and addresses the computational cost concerns raised by the reviewers by including analyses of the method's efficiency.

**Reviewer Scores:**

After rebuttal, novelty and positioning concerns are not fully addressed, and therefore the AC would not expect substantial changes in the ratings given by the reviewers. Although the submission received one positive rating of 8, the reviewer's written comments do not provide enough support to overweigh the remaining concerns.

---

### Decision · Program_Chairs · 2026-01-26

Reject